# Polyphonia: Zero-Shot Timbre Transfer in Polyphonic Music with Acoustic-Informed Attention Calibration

**Haowen Li** [* 1]  **Tianxiang Li** [* 1]  **Yi Yang** [1]  **Boyu Cao** [1]  **Qi Liu** [† 1]

## Abstract

The advancement of diffusion-based text-to-music generation has opened new avenues for zero-shot music editing. However, existing methods fail to achieve stem-specific timbre transfer, which requires altering specific stems while strictly preserving the background accompaniment. This limitation severely hinders practical application, since real-world production necessitates precise manipulation of components within dense mixtures. Our key finding is that, while vanilla cross-attention captures semantic features of stems, it lacks the spectral resolution to strictly localize targets in dense mixtures, leading to boundary leakage. To resolve this dilemma, we propose *Polyphonia*, a zero-shot editing framework with Acoustic-Informed Attention Calibration. Rather than relying solely on diffuse semantic attention, Polyphonia leverages a probabilistic acoustic prior to establish coarse boundaries, enabling non-target stems preserved precise semantic synthesis. For evaluation, we propose *PolyEvalPrompts*, a standardized prompt set with 1,170 timbre transfer tasks in polyphonic music. Specifically, *Polyphonia* achieves an increase of 15.5% in target alignment compared to baselines, while maintaining competitive music fidelity and non-target integrity.

## 1. Introduction

The landscape of generative music has been revolutionized by Diffusion Models (Ho et al., 2020; Liu et al., 2024b; Schneider et al., 2023; Ghosal et al., 2023; Evans et al., 2025), enabling the synthesis of high-fidelity music from textual descriptions. However, the transition from novelty

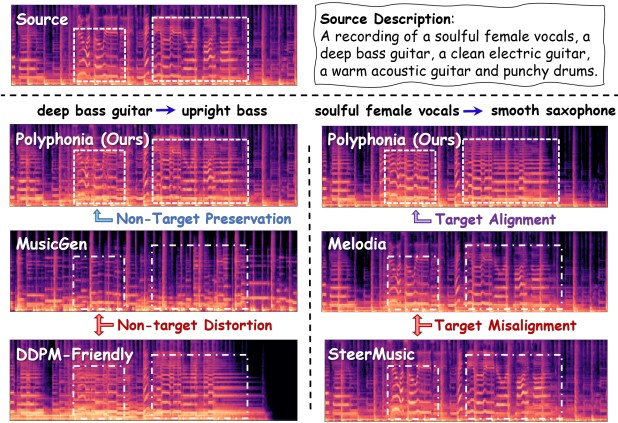

*Figure 1.* Illustration of stem-specific timbre transfer with Polyphonia compared against baselines on two tasks. While MusicGen and DDPM-Friendly suffer from *Non-target Distortion* where the vocals are distorted, Polyphonia achieves robust *Non-Target Preservation*, strictly preserving the background rhythm. The right panel displays a vocal-to-saxophone transfer. Furthermore, Melodia and SteerMusic fail to disentangle the target from the accompaniment, resulting in *Target Misalignment*. In contrast, Polyphonia successfully synthesizes the desired timbre with precise *Target Alignment*.

tools to professional music production assistants necessitates more than just generation; surgical editing controllability is paramount. In professional workflows, text-driven music editing generally addresses two levels of manipulation: *inter-stem editing* (e.g., adding or removing a specific instrument) and *intra-stem editing* (e.g., replacing a specific instrument). Within the latter, *stem-specific timbre transfer* represents a critical yet challenging task: surgically altering the timbre of a target stem (e.g., transforming a vocals track into a violin) while strictly preserving the acoustic integrity of its original pitch contours, rhythmic structure and all non-target stems (e.g., the co-occurring melody and accompaniment). This task poses a dual challenge: the newly transferred timbre must accurately reflect the semantic prompt while seamlessly blending with the rigidly preserved non-target stems.

While zero-shot editing paradigms have shown promise, applying them to multi-track music mixtures is impeded by severe spectral interference. As illustrated in Fig. 1, existing approaches generally succumb to a dilemma rooted

---

[*]Equal contribution [1]School of Future Technology, South China University of Technology, Guangzhou, China.. Correspondence to: Qi Liu <drliuqi@scut.edu.cn>.

*Proceedings of the 43rd International Conference on Machine Learning*, Seoul, South Korea. PMLR 306, 2026. Copyright 2026 by the author(s).

in their attention mechanisms. The first category (Copet et al., 2023; Manor & Michaeli, 2024; Meng et al.) relies on vanilla cross-attention. While effective at capturing semantic features, it struggles to confine the generation within precise spectral boundaries in dense mixtures. This leads to boundary leakage, causing the non-target stems to be regenerated alongside the target. To mitigate this, the second category (Yang et al., 2026; Niu et al., 2026; Zhang et al., 2024) introduces rigid preservation constraints via internal feature preservation. However, in dense mixtures, these strict constraints indiscriminately preserve entangled features, conflicting with the editing objective and preventing the target from manifesting. Essentially, current models lack an acoustic boundary to distinguish the "to-be-edited" content from the "to-be-preserved" non-target stems.

This diagnosis points to a crucial conclusion: to achieve precise stem-specific timbre transfer, the localization paradigm needs to incorporate acoustic prior to constrain the diffuse nature of semantic attention. To realize this, we propose *Polyphonia*, a zero-shot framework designed to tackle this semantic-acoustic misalignment, resolving the mismatch where high-level semantic attention fails to align with the fine-grained precision of acoustic energy distribution. Surpassing naive separation-remixing strategies and naive magnitude masks, Polyphonia rigorously formulates a probabilistic acoustic prior called Ideal Ratio Mask (IRM) (Wang et al., 2014; Narayanan & Wang, 2013) by analyzing the spectral energy competition within the mixture via Blind Source Separation (BSS). Subsequently, we introduce *Acoustic-Informed Attention Calibration* to leverage the acoustic prior to selectively preserve background features while aligning generative focus to the target, enforcing precise spectral editing control.

To standardize the evaluation of stem-specific timbre transfer, we construct *PolyEvalPrompts*, a comprehensive prompt set applied to the MUSDB18-HQ (Rafii et al., 2019) test subset and MusicDelta (Bittner et al., 2014) datasets. Extensive experiments demonstrate that Polyphonia outperforms state-of-the-art (SOTA) baselines, achieving the highest textual alignment (CLAP) and an optimal balance between editing strength and non-target integrity.

Our contributions are summarized as follows:

- We formalize the *Semantic-Acoustic Misalignment* as a fundamental barrier, demonstrating that semantic attention inherently lacks the spectral resolution to localize targets in dense mixtures.

- We propose *Polyphonia*, a zero-shot framework that introduces *Acoustic-Informed Attention Calibration*. By injecting a rigorously formulated probabilistic acoustic prior, it solves the semantic-acoustic misalignment, strictly preserving the non-target stems while enforcing

target semantics.

- We contribute *PolyEvalPrompts*, a standardized prompt set with 1,170 editing tasks applied to MUSDB18-HQ and MusicDelta, establishing a reproducible testbed for assessing stem-specific timbre transfer.

## 2. Related Work

### 2.1. Text-to-Music Generation

The domain of music generation has been reshaped by autoregressive transformers (Copet et al., 2023) and Latent Diffusion Models (LDMs) (Liu et al., 2023; 2024b; Schneider et al., 2023; Ghosal et al., 2023). While models like AudioLDM 2 (Liu et al., 2024b) and Stable Audio (Evans et al., 2025) achieve high-fidelity holistic generation, they fundamentally lack surgical controllability over individual acoustic components. To address this, recent research has resorted to supervised fine-tuning. For instance, Music ControlNet (Wu et al., 2024) adapts the ControlNet architecture to music, requiring extensive retraining to inject specific spatially or temporally dense controls. Similarly, methods like Instruct-MusicGen (Zhang et al., 2025) and Jen-1 Composer (Yao et al., 2025b) employ extensive instruction tuning to enable editing capabilities. However, these supervised paradigms incur prohibitive computational costs, require massive paired datasets during training.

### 2.2. Zero-Shot Text-to-Music Editing

To bypass fine-tuning costs, research has pivoted toward zero-shot editing pipelines. Early approaches adapted global processing techniques like SDEdit (Meng et al.) and DDIM/DDPM Inversion (Song et al.; Manor & Michaeli, 2024) to music. However, operating on the entire latent space without localization often leads to structural collapse or non-target distortion, where the background accompaniment is inadvertently altered (Copet et al., 2023). To improve preservation, recent works have sought to impose constraints via internal feature manipulation. Approaches such as Melodia (Yang et al., 2026), MEDIC (Liu et al., 2024a), and PPAE (Xu et al., 2024) extend attention manipulation paradigms successful in vision (Hertz et al.; Tumanyan et al., 2023; Cao et al., 2023) to the music domain. Notably, PPAE is primarily optimized for general audio events with sparse spectral layouts; in contrast, our task focuses on highly entangled polyphonic music, where targets are deeply embedded and spectrally overlapped within dense mixtures. Others like SteerMusic (Niu et al., 2026) and MusicMagus (Zhang et al., 2024) employ gradient-based energy guidance. Nevertheless, these frameworks fundamentally rely on internal representations (e.g., self/cross-attention maps or energy gradients) to maintain coherence. We argue that in dense multi-track mixtures, these internal features are

inherently contaminated by spectral interference, rendering them insufficient for precise disentanglement.

### 2.3. Audio-Guided Masking Paradigms

The utilization of audio cues to construct spatial masks has been predominantly explored in Audio-Visual Segmentation (AVS) (Zhou et al., 2022; 2025), where auditory signals guide the pixel-level localization of sounding objects in video frames. While AVS relies on a cross-modal, discriminative assumption to map audio semantics to mutually exclusive visual pixels, our task operates in a fundamentally different intra-modal, generative setting. Rather than segmenting distinct visual objects, *Polyphonia* utilizes acoustic priors as continuous structural boundaries within the latent space to disentangle dense, spectrally interfering audio mixtures.

## 3. Preliminary

**Attention Mechanism** (Vaswani et al., 2017) was introduced to selectively aggregate contextual features:

$$E = \frac{QK^\top}{\sqrt{d}} \tag{1}$$

$$\text{Attn}(Q, K, V) = \text{Softmax}(E)V = AV \tag{2}$$

where $E$ is the attention energy matrix and $A$ is the attention map.

**AudioLDM 2** (Liu et al., 2024b), a conditional latent diffusion model (LDM) tailored for music generation, serves as the backbone of our framework. Specifically, AudioLDM 2 operates in the time-frequency domain. It utilizes a Variational Autoencoder (VAE) to compress the input Log-Mel Spectrogram $X \in \mathbb{R}^{T \times F}$ into a low-dimensional latent code $z = \mathcal{E}(X)$. A 16-layer T-UNet processes the hidden features $\varphi(z)$ using interleaved Self-Attention (SA) and Cross-Attention (CA) blocks. SA layers, which model global temporal dependencies, query, key, and value are all derived from the music features: $Q, K, V = W_{\{Q,K,V\}} \cdot \varphi(z)$. Conversely, CA layers inject external control; while $Q$ remains derived from $\varphi(z)$, $K$ and $V$ are projected from the condition embeddings $\mathcal{C} \in \{\mathcal{C}_{\text{text}}, \mathcal{C}_{\text{LoA}}\}$, formulated as $K, V = W_{\{K,V\}} \cdot \mathcal{C}$. This distinguishes AudioLDM 2 from standard LDMs (Rombach et al., 2022) that rely solely on text embeddings. As a hierarchical hybrid conditioning mechanism, it synergizes the *Text Condition* ($\mathcal{C}_{\text{text}}$), which extracts linguistic semantics via T5 (Chung et al., 2024) and CLAP (Elizalde et al., 2023) encoders, with the *Language of Audio (LoA) Condition* ($\mathcal{C}_{\text{LoA}}$), a sequence of GPT-2 (Radford et al., 2019) generated acoustic style representations to encapsulate fine-grained acoustic details.

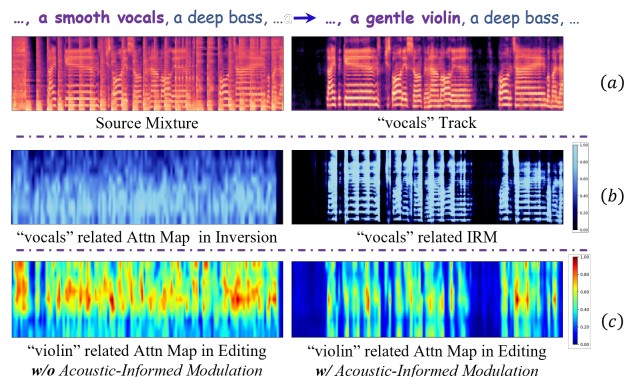

*Figure 2.* Illustration of the Semantic-Acoustic Misalignment. (a) Spectral Interference: The input mixture (left) and the corresponding vocals track (right), illustrating the challenge where stems spatially overlap. (b) Semantic vs. Acoustic: During inversion, the text cross-attention map (left) exhibits a diffuse distribution, failing to pinpoint the vocals. In contrast, the Ideal Ratio Mask (IRM) (right) serves as a sharp, continuous acoustic prior, accurately quantifying vocals' energy dominance. (c) Effect of Modulation: When editing the stem from "vocals" to "violin", the naive attention map (left) suffers from severe leakage due to semantic misalignment. By utilizing the acoustic prior, our Acoustic-Informed Modulation (right) strictly constrains the attention to the original vocal spectral envelope, ensuring precise timbre transfer. *Attention maps are visualized by averaging weights across heads and time steps from the first layer of the T-UNet.*

## 4. Method

Given a log-mel spectrogram $X_0$ of a multi-track mixture, we can define $X_0$ as a fusion $\mathcal{F}$ of linear complex spectrograms of the target stem $S_{\text{tgt}}$ and non-target mixture $S_{\text{con}}$ by applying Mel-filterbank transformation $\mathcal{M}$:

$$X_0 = \mathcal{F}(S_{\text{tgt}}, S_{\text{con}}) = \log\left(\mathcal{M}(|S_{\text{mix}}|) + \epsilon\right),$$
$$|S_{\text{mix}}|^2 = |S_{\text{tgt}}|^2 + |S_{\text{con}}|^2 + 2|S_{\text{tgt}}||S_{\text{con}}|\cos(\Delta\phi) \tag{3}$$

where $\Delta\phi$ denotes the phase difference and $\epsilon$ is a small scalar avoiding $\log(0)$. Given a target prompt $Y_{\text{tgt}}$, to achieve *stem-specific timbre transfer* in a zero-shot manner, the editing process $\mathcal{P}(X_0, Y_{\text{tgt}}|\theta)$ generating a new mixture $\hat{X}_0$ within pre-trained parameters $\theta$ can be formulated as:

$$\hat{X}_0 = \mathcal{P}(X_0, Y_{\text{tgt}}|\theta) = \mathcal{F}(\hat{S}_{\text{tgt}}, S_{\text{con}}) \tag{4}$$

Consequently, generating $\hat{X}_0$ requires seamless integration where the new target stem $\hat{S}_{\text{tgt}}$ respects the acoustic coherence of the preserved non-target mixture $S_{\text{con}}$.

While previous studies (Yang et al., 2026; Zhang et al., 2024; Niu et al., 2026; Manor & Michaeli, 2024) fundamentally predicate their success on the localization capacity of vanilla cross-attention. However, as visualized in Fig. 2 (c) (left), this assumption collapses in multi-track mixtures due to the fundamental physical disparity between visual and auditory data: Spectral Interference.

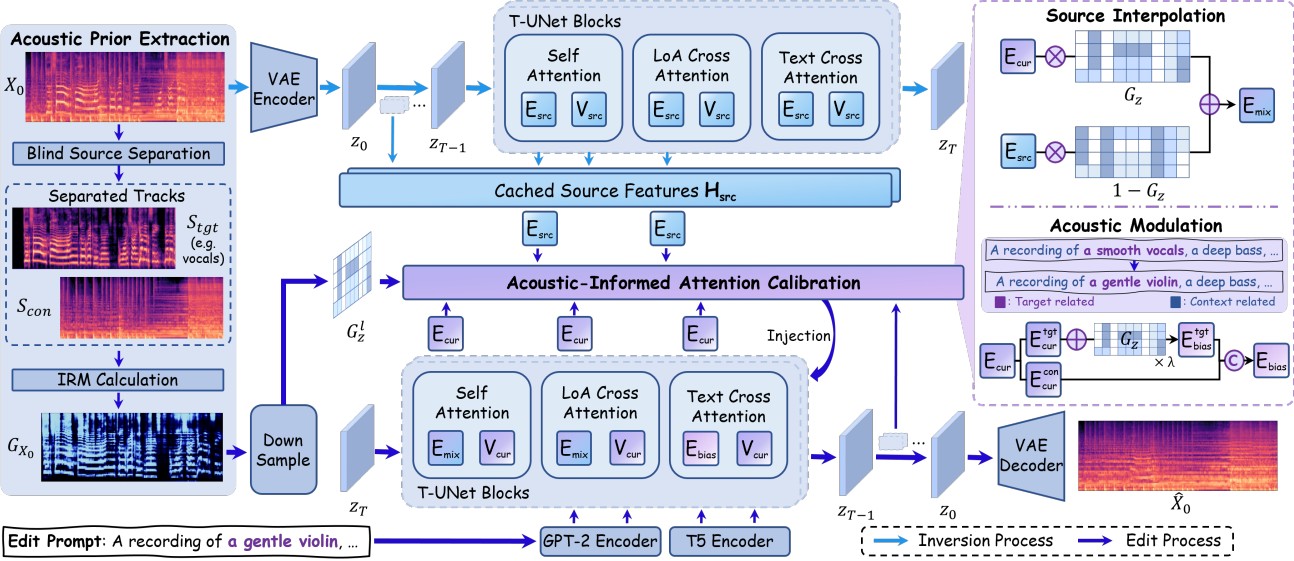

*Figure 3.* **Overview of Polyphonia.** The pipeline follows a dual-path mechanism: Inversion and Editing. Given an input music $X_0$ and an edit prompt $Y_{tgt}$, it first performs *Acoustic Prior Extraction* to obtain $G_{X_0}$. During the Inversion Process, the input is encoded into latent space, and source features $\mathcal{H}(X_0)$ are cached from the T-UNet blocks. In the Edit Process, it introduces the Acoustic-Informed Attention Calibration module to regulate the generation via two mechanisms: (1) Source Interpolation. It blends $\mathcal{H}(X_0)$ with current features guided by the acoustic prior $G$ to preserve background consistency; (2) Acoustic Modulation. It utilizes $G$ as a layout bias. Specifically, $G$ is scaled and added to the target-related energy matrix to enforce the target alignment, before being concatenated with context features to form the final energy matrix $E_{bias}$. The complete editing procedure is summarized in **Algorithm 1**.

To elucidate this, consider the formulation of the input data $X$. In the image domain, a matrix of the visual scene $\mathbf{X}^{img} \in \mathbb{R}^{H \times W}$ is typically made up of opaque objects:

$$\forall i \in [1,H], j \in [1,W], \quad \mathbf{M}_{i,j} \in \{0,1\}$$
$$\mathbf{X}^{img}_{i,j} = \mathbf{M}_{i,j} \cdot \mathbf{O}^{tgt}_{i,j} + (1 - \mathbf{M}_{i,j}) \cdot \mathbf{O}^{con}_{i,j} \quad (5)$$

where the pixel value $\mathbf{X}^{img}_{i,j}$ for a mixture of a target $\mathbf{O}^{tgt}_{i,j}$ and background $\mathbf{O}^{con}_{i,j}$ is governed by a binary occlusion mask $\mathbf{M}$. This implies a principle of mutual exclusivity: a pixel belongs to either the target or the non-target stems, but rarely both. Consequently, the latent query vector $Q$ derived from the encoder represents a single distinct semantic entity, allowing cross-attention to easily disentangle sources based on spatial activation during the editing process.

In contrast, the audio domain is governed by spectral interference. As shown in Eq. 4, a matrix of music mixture $\mathbf{X}^{music} \in \mathbb{R}^{T \times F}$ aggregates sources simultaneously:

$$\forall i \in [1,T], j \in [1,F], \quad \mathbf{X}^{music}_{i,j} = \mathcal{F}(\mathbf{S}^{tgt}_{i,j}, \mathbf{S}^{con}_{i,j}) \quad (6)$$

Unlike the image domain, there is no binary mask $\mathbf{M}$ that spatially separates the sources; instead, the input inherently encodes a coupled representation. Consequently, the latent-derived Query $Q$ encapsulates mixed features rather than discrete objects. When calculating CA in the editing process, this ambiguity causes the query to correlate indiscriminately with both target and non-target keys, leading to

the non-target distortion visualization in Figure 2 (c) (left). This creates a critical bottleneck: unlike single-track editing where targets are isolated, in dense mixtures, such unconstrained guidance conflicts with rigid structural constraints, which results in the Target Misalignment visualized in Fig. 1 (right).

To tackle this semantic-acoustic misalignment where specific semantic features is undermined by ambiguous spatial guidance, our investigation addresses two fundamental challenges: (1) Given that internal cross-attention fails to disentangle the target from the mixture, how can we acquire a precise target spectral envelope to rectify the model's focus? (2) How can we calibrate the diffusion features using this profile to enforce the target alignment while strictly preserving the non-target integrity?

To address these challenges, we propose *Polyphonia*. As illustrated in Fig. 3, our framework operates in a dual-path manner involving Inversion (Song et al.) and Editing. Specifically, our pipeline consists of two core components designed to rectify semantic-acoustic misalignment: (1) **Acoustic Prior Extraction** (Sect. 4.1): We first decompose the mixture $X_0$ to derive a probabilistic acoustic prior $G_{X_0}$, providing an acoustic reference for precise editing. (2) **Acoustic-Informed Attention Calibration** (Sect. 4.2): During the editing process, we employ Acoustic-Informed Attention Calibration to inject this prior into diffusion features, ensuring precise targets localization and non-targets

preservation.

## 4.1. Acoustic Prior via Ideal Ratio Mask

To answer the first challenge, we need an objective reference that quantifies the target profile correctly. Our investigation begins by evaluating the reliability of internal attention before resorting to external acoustic knowledge.

### 4.1.1. INTERNAL PRIOR FAILURE

While image editing approaches (Couairon et al.; Cao et al., 2023; Simsar et al., 2025) and PPAE (Xu et al., 2024) leverage CA maps from inversion as priors, this strategy is ineffective in the dense mixture where the aforementioned spectral interference results in diffusely distributed maps. As evidenced in Fig. 2 (b) (left), even conditioned on the correct source prompt, the internal attention fails to distinguish the target stem from the non-target stems, necessitating an external reference to ground the localization.

### 4.1.2. EXTERNAL ACOUSTIC PRIOR

To derive the acoustic reference, one might initially consider relying on basic acoustic descriptors such as pitch tracks or energy envelopes (Wu et al., 2024; Melechovsky et al., 2024). However, these features offer partial guidance, and fail to distinguish the target from the non-target stems. To explicitly counteract spectral interference in a zero-shot manner, we instead leverage Blind Source Separation (BSS) (Rouard et al., 2023; Lu et al., 2024), which decomposes the mixture to the estimated target stem $\tilde{S}_{\text{tgt}}$ and non-target mixture $\tilde{S}_{\text{con}}$ that the attention failed to capture. Crucially, for instruments that do not fall into the primary categories of the BSS model (e.g., mapping melodic instruments like *piano* or *guitar* to the generic "Others" category), we employ a target-to-stem mapping strategy; detailed taxonomic configurations are provided in **App. C.1**.

Given the decomposed stems $\tilde{S}_{\text{tgt}}$ and $\tilde{S}_{\text{con}}$, a naive alternative would be a "separation-editing-remixing" pipeline. However, such a pipeline suffers from a fundamental *Contextual Mismatch*: the independent generation of $\hat{S}_{\text{tgt}}$ is blind to the acoustic environment of the non-target stems. Consequently, even if remixed via precise waveform superposition, the resulting mixture lacks coherence, leading to artifacts where the target sounds detached from the accompaniment. As demonstrated in Fig. 4, holistic editing outperforms such baselines, achieving superior spectral integrity and higher SongEval-based coherence scores (Yao et al., 2025a).

Since independent generation fails to preserve the acoustic unity, we turn to extract a soft attention guidance $G$, which allows the diffusion model to synthesize coherent transitions from $X_0$ to $\hat{X}_0$. By approximating the interference $\mathcal{F}$ via this prior, we instantiate editing process $\mathcal{P}$ as a vision-like probabilistically masked feature fusion:

$$\mathcal{P}(X_0, Y_{\text{tgt}}|\theta) \coloneqq \underbrace{G \odot \mathcal{G}(Y_{\text{tgt}})}_{\text{Semantic Injection}} + \underbrace{(1 - G) \odot \mathcal{H}(X_0)}_{\text{Non-Target Preservation}} \quad (7)$$

where $\mathcal{G}(Y_{\text{tgt}})$ denotes the generative features conditioned on the target, and $\mathcal{H}(X_0)$ is the cached hidden features of $X_0$. Unlike binary masks $\mathbf{M} \in \{0, 1\}$ in vision, $G \in [0, 1]$ functions as a continuous spectral soft mask. This effectively linearizes the editing task: high-probability regions synthesize the target timbre, while low-probability regions rigidly preserve the non-target stems $S_{\text{con}}$.

To quantify $G$, a naive approach is define $G = G_{\text{norm}} = \mathcal{N}(|\tilde{S}_{\text{tgt}}|)$, where $\mathcal{N}$ denotes Min-Max normalization. However, this metric is loudness-based rather than probability-based; it ignores the background accompaniment, leading to non-target distortion in mixtures. To address this, we define our acoustic prior using the Ideal Ratio Mask (IRM) (Wang et al., 2014; Narayanan & Wang, 2013), which incorporates the non-target estimate $\tilde{S}_{\text{con}}$ to model the energetic competition:

$$G_{\text{IRM}} = \sqrt{\frac{|\tilde{S}_{\text{tgt}}|^2}{|\tilde{S}_{\text{tgt}}|^2 + |\tilde{S}_{\text{con}}|^2}} \quad (8)$$

Unlike $G_{\text{norm}}$, $G_{\text{IRM}}$ represents the *spectral probability* of the target's dominance. It naturally suppresses the guidance in regions where the background energy prevails, ensuring that edits only occur where the target is perceptually salient. As visually corroborated in Fig. 2, the raw target stem (approximating $G_{\text{norm}}$) in (a, right) shows continuous activity, while $G_{\text{IRM}}$ in (b, right) exhibits distinct suppression gaps where non-target stems dominates. Finally, to align this acoustic prior with AudioLDM 2's input space, we apply the Mel-filterbank transformation $\mathcal{M}$:

$$G_{X_0} = \sqrt{\frac{\mathcal{M}(|\tilde{S}_{\text{tgt}}|^2)}{\mathcal{M}(|\tilde{S}_{\text{tgt}}|^2) + \mathcal{M}(|\tilde{S}_{\text{con}}|^2)}} \quad (9)$$

Crucially, we retain these continuous soft values to respect the superposition nature of audio. For diffusion guidance, $G_{X_0}$ is downsampled to match the resolution of LDM layer $l$, denoted as $G_z^l$. Finally, $G = G_z^l$ is the acoustic prior.

## 4.2. Acoustic-Informed Attention Calibration

We posit that localization should not rely on a single modality. Specifically, acoustic prior $G$ establishes the coarse acoustic boundary (suppressing irrelevant energy like drums), while the cross-attention mechanism performs fine-grained semantic selection within this permissible region. Consequently, we introduce a dual-calibration leveraging $G$: Source Interpolation to preserve non-target stems via $\mathcal{H}(X_0)$, and Acoustic Modulation to align target generation with acoustic prior and target prompt $Y_{\text{tgt}}$.

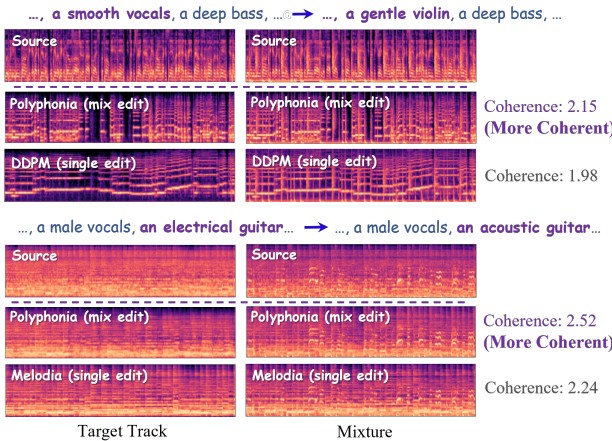

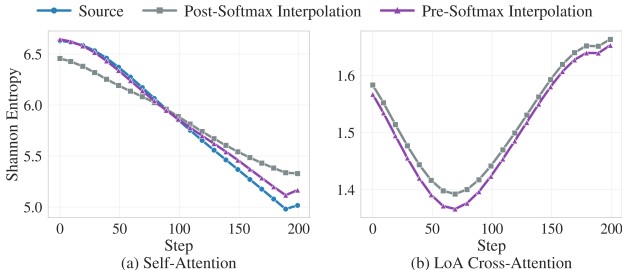

*Figure 5.* Shannon Entropy analysis of attention maps across steps. (a) SA: Pre-Softmax Interpolation closely tracks the Source, indicating superior structure preservation. (b) LoA CA: Pre-Softmax Interpolation achieves lower entropy, reflecting a sharper distribution critical for precise localization. For more details of this analysis, please refer to **App. F**.

*Figure 4.* Quality Comparison of acoustic coherence between holistic mixture editing and separation-based single-track editing. Coherence $\in [0, 5]$ *is measured by SongEval* (Yao et al., 2025a).

### 4.2.1. SOURCE INTERPOLATION

To strictly enforce background consistency, Polyphonia extends the injection paradigm (Yang et al., 2026) to both Self-Attention and LoA Cross-Attention. Since LoA is conceptualized as global acoustic textures akin to latent features $\phi(z_t)$, it requires rigid preservation akin to Self-Attention.

To ensure this structural preservation is precise, Selective Pre-Softmax Interpolation is implemented. Distinct from post-Softmax methods (Cao et al., 2023; Hertz et al.) that linearly smear probabilities, it operates in the raw logit space leverages the non-linear amplification of the Softmax function. We formulate the calibrated attention as:

$$E_{\text{mix}} = (1 - G) \odot E_{\text{src}} + G \odot \frac{QK^\top}{\sqrt{d}} \quad (10)$$

$$\text{Attn}_{\text{itp}}(Q, K, V) = \text{Softmax}(E_{\text{mix}}) V \quad (11)$$

where $E_{\text{src}} \in \mathcal{H}(X_0)$ is the cached attention energy matrix. As shown in Fig. 5, for LoA, this mechanism crystallizes diffuse textures into a sharper structural foundation (reduced entropy); for SA, it strictly maintains fidelity to the source's sparsity pattern, avoiding the artificial entropy increase induced by linear probability averaging. Detailed implementation regarding the tensor alignment and broadcasting mechanisms of $G$ is provided in **App. C.4**.

### 4.2.2. ACOUSTIC MODULATION

While Source Interpolation stabilizes the non-target structure, synthesizing distinct target semantics necessitates explicit spatial grounding. Vanilla cross-attention often yields diffuse activation maps lacking spatial-textual binding. To rectify this, Acoustic Modulation injects the prior $G$ as an inductive bias directly into the text cross-attention logits.

The process begins by constructing a binary token mask

$\mathbf{m}^{\text{text}} \in \{0, 1\}^{L_y}$ to isolate the target entity within the prompt $Y_{\text{tgt}}$. Let $\mathcal{I}_{\text{tgt}}$ denote the set of token indices of the target subject (e.g., "violin"). The mask is defined as:

$$\mathbf{m}_i^{\text{text}} = \begin{cases} 1, & \text{if } i \in \mathcal{I}_{\text{tgt}} \\ 0, & \text{otherwise} \end{cases} \quad (12)$$

Subsequently, a Spatio-Textual Bias Matrix $\mathbf{B}$ is synthesized via the outer product of $\mathbf{g} = \text{Flatten}(G) \in \mathbb{R}^{L_z}$ and the token mask $\mathbf{m}^{\text{text}}$. This bias is injected into the pre-Softmax attention energy to regulate the attention distribution:

$$E_{\text{bias}} = \frac{QK^\top}{\sqrt{d}} + \lambda \cdot \mathbf{B}, \quad \mathbf{B} = \mathbf{g} \otimes \mathbf{m}^{\text{text}} \in \mathbb{R}^{L_z \times L_y} \quad (13)$$

$$\text{Attn}_{\text{mod}}(Q, K, V) = \text{Softmax}(E_{\text{bias}}) V \quad (14)$$

where $\lambda$ is a scalar controlling modulation strength. This operation effectively reshapes the attention landscape. As visualized in Fig. 2, by selectively boosting energy potentials at the intersection of high acoustic probability and target semantic tokens, the mechanism forces the generative focus to strictly align with the target's spectral envelope, thereby eliminating semantic leakage. Detailed implementation regarding the tensor alignment and broadcasting mechanisms of $G$ is provided in **App. C.4**.

## 5. Experiments

### 5.1. Experimental Setup

**Dataset.** Evaluations are conducted on two multi-track benchmarks: the **MUSDB18-HQ** (Rafii et al., 2019) test subset (50 full-track songs), serving as the high-fidelity standard for source separation; and the **MusicDelta** subset of MedleyDB (Bittner et al., 2014) (28 mixtures), selected for its diversity in genre and instrumentation. These benchmarks cover a wide range of track complexities from 1 to 9 stems, providing a robust basis for evaluating algorithmic

*Table 1.* **Objective and Subjective Quantitative comparison on MusicDelta and Musdb18-HQ datasets.** First , Second , and Third indicate the best, second best, and third best performance respectively.

| Method | | Objective Evaluation | | | | | | | Subjective Evaluation | | |
|---|---|---|---|---|---|---|---|---|---|---|---|
| | | CLAP↑ | CQT1-PCC↑ | LPAPS↓ | FAD↓ | KAD↓ | ASB↑ | AMB↑ | TTA↑ | CTI↑ | GAC↑ |
| **MusicDelta** | SDEdit | 0.119±0.126 | 0.090±0.228 | 6.907±0.667 | 1.914±0.177 | 0.942±0.175 | 0.000 | 0.000 | 1.125±0.765 | 1.554±0.876 | 1.458±0.798 |
| | DDIM | 0.353±0.119 | 0.253±0.216 | 5.586±0.662 | 1.155±0.242 | 0.782±0.230 | 0.512 | 0.500 | 2.753±1.012 | 2.305±0.987 | 3.084±1.023 |
| | DDPM | 0.351±0.117 | 0.274±0.217 | 5.490±0.683 | 1.069±0.217 | 0.765±0.228 | 0.534 | 0.533 | 2.792±1.034 | 2.257±0.992 | 3.026±1.045 |
| | Melodia | 0.380±0.104 | 0.513±0.220 | 3.540±0.509 | 0.715±0.187 | 0.627±0.264 | 0.903 | 0.864 | 3.215±1.023 | 3.594±0.895 | 3.467±1.012 |
| | SteerMusic | 0.317±0.114 | 0.556±0.197 | 3.614±0.597 | 0.738±0.178 | 0.607±0.257 | 0.761 | 0.767 | 3.156±1.054 | 3.435±0.976 | 3.318±1.089 |
| | MusicMagus | 0.238±0.113 | 0.361±0.218 | 4.690±0.585 | 1.192±0.222 | 0.769±0.220 | 0.479 | 0.462 | 2.364±0.932 | 3.116±0.954 | 2.745±0.965 |
| | MusicGen | 0.377±0.092 | 0.069±0.217 | 6.142±0.573 | 1.331±0.284 | 0.788±0.188 | 0.355 | 0.000 | 3.592±0.956 | 2.054±0.843 | 3.623±0.987 |
| | **Polyphonia** | **0.437±0.101** | 0.547±0.225 | 4.096±0.643 | 0.949±0.233 | 0.695±0.239 | **0.910** | **0.991** | **3.804±0.912** | 3.413±0.945 | **3.692±0.928** |
| **MUSDB18-HQ test** | SDEdit | 0.093±0.106 | 0.031±0.183 | 7.118±0.535 | 1.835±0.127 | 0.889±0.104 | 0.000 | 0.000 | 1.082±0.754 | 1.365±0.843 | 1.393±0.789 |
| | DDIM | 0.277±0.110 | 0.225±0.211 | 6.011±0.493 | 1.199±0.211 | 0.720±0.142 | 0.469 | 0.653 | 2.627±1.043 | 2.165±1.012 | 2.903±1.056 |
| | DDPM | 0.283±0.101 | 0.243±0.209 | 5.842±0.470 | 1.084±0.165 | 0.684±0.140 | 0.521 | 0.691 | 2.664±1.065 | 2.112±1.023 | 2.876±1.078 |
| | Melodia | 0.296±0.103 | 0.363±0.246 | 3.893±0.387 | 0.655±0.149 | 0.495±0.182 | 0.898 | 0.877 | 3.085±1.067 | 3.523±0.902 | 3.386±1.034 |
| | SteerMusic | 0.255±0.116 | 0.383±0.245 | 4.105±0.487 | 0.747±0.216 | 0.497±0.186 | 0.767 | 0.788 | 2.953±1.087 | 3.345±0.998 | 3.234±1.102 |
| | MusicMagus | 0.187±0.101 | 0.282±0.218 | 5.016±0.419 | 1.186±0.167 | 0.711±0.142 | 0.476 | 0.496 | 2.186±0.965 | 3.054±0.987 | 2.625±0.996 |
| | MusicGen | 0.295±0.090 | 0.003±0.135 | 6.600±0.308 | 1.374±0.180 | 0.840±0.125 | 0.268 | 0.000 | 3.342±0.978 | 1.906±0.856 | 3.445±0.989 |
| | **Polyphonia** | **0.342±0.094** | 0.371±0.231 | 4.426±0.550 | 0.868±0.202 | 0.645±0.199 | **0.910** | **0.985** | **3.764±0.934** | 3.395±0.967 | **3.512±0.945** |

performance (detailed track count distributions are provided in **App. H**).

*PolyEvalPrompts*. Existing prompt sets often lack the granularity required to evaluate stem-specific timbre transfer, where specific instrumental attributes must be targeted without perturbing the accompaniment. To address this, we construct PolyEvalPrompts tailored to these datasets, using a two-stage pipeline:

1) *Acoustic Analysis:* We leverage the Qwen-Audio (Chu et al., 2023) to analyze the acoustic content of each mixture, generating comprehensive metadata. It describes the *instrumentation* (e.g., distorted electric guitar), *playing techniques*, and the overall *musical style*, ensuring that the source description is semantically rich and accurate.

2) *Task Synthesis:* Conditioned on the extracted metadata, we systematically define 15 distinct editing tasks for each musical piece within Qwen3 (Yang et al., 2025), spanning a spectrum from subtle timbre transfers to radical instrument timbre style alterations. Pipeline details and prompt samples are provided in **App. I**.

**Baselines.** We benchmark Polyphonia against a diverse set of SOTA editing paradigms: (1) **Global Inversion Methods**: SDEdit (Meng et al.), DDIM Inversion (Song et al.), and DDPM-Friendly (Manor & Michaeli, 2024), which apply diffusion inversion and denoising on the entire latent space; (2) **Structural Steering Methods**: Melodia (Yang et al., 2026), which employs self-attention injection, along with SteerMusic (Niu et al., 2026) and MusicMagus (Zhang et al., 2024), which utilize gradient-based energy guidance to steer the generation; and (3) **Pre-trained Autoregressive Models**: MusicGen (Copet et al., 2023) (using the `melody-medium` checkpoint), which serves as an

upper bound for structural conditioning. Note that while PPAE (Xu et al., 2024) is relevant, it is excluded from the quantitative benchmark due to reproduction challenges and the absence of official hyperparameter configurations. Detailed settings of baselines are provided in **App. G**.

**Objective Metrics.** To quantify target alignment, we utilize the **CLAP** score (Wu et al., 2023). For structural and melodic preservation, we employ **LPAPS** (Paissan et al., 2024) (for temporal coherence) and **CQT1-PCC** (Brown, 1991; Niu et al., 2026) (for core melodic consistency). To assess distributional consistency to source music, we adopt **Kernel Audio Distance (KAD)** (Chung et al., 2025), an unbiased alternative to **Fréchet Audio Distance** (FAD) (Kilgour et al., 2019). Furthermore, composite metrics including **Adherence-Structure Balance (ASB)** and **Adherence-Musicality Balance (AMB)** (Yang et al., 2026) are introduced, which calculate the harmonic mean of the normalized alignment score (CLAP) and the respective preservation scores (LPAPS for ASB, CQT1-PCC for AMB), ensuring a holistic assessment of trade-off. Details in **App. J**.

**Subjective Metrics.** Complementing objective analysis, we conducted a Mean Opinion Score (MOS) study involving human raters on a 5-point Likert scale. Three defined perceptual dimensions specifically mapped to stem-specific timbre transfer challenges: (1) **Target Timbre Alignment (TTA)** measuring target fidelity; (2) **Contextual Timbre Integrity (CTI)** evaluating non-target integrity; (3) **Global Acoustic Coherence (GAC)** assessing the spectral blend. **Detailed definitions are provided in App. K**.

**Implementation Details.** Polyphonia is implemented via the pre-trained **AudioLDM 2** (Liu et al., 2024b) and **HT-Demucs** (Rouard et al., 2023) (BSS model) on a single NVIDIA GeForce RTX 3090 GPU. More details of imple-

mentation are in **App. C**. Editing process goes through 100 diffusion steps. **Classifier-Free Guidance (CFG) (Ho & Salimans) for target alignment is set to 3.5**. For Acoustic Modulation, the **scalar** $\lambda$ **is set to 2.5**. The calibration is applied to all down layers of T-UNet. Detailed analysis of these choices is provided in **Sect. 5.4 and App. E**. Efficiency analysis of Polyphonia is provided in **App. D**.

## 5.2. Objective Evaluation

We quantitatively and qualitatively evaluate Polyphonia against baselines on the MusicDelta and MUSDB18-HQ test datasets via PolyEvalPrompts. As summarized in Tab. 1, existing paradigms struggle to balance target modification with background fidelity. Methods without structural steering, such as MusicGen and DDPM-Friendly, exhibit low LPAPS and CQT1-PCC, indicating a failure to preserve background. Conversely, structural steering approaches like Melodia and Steermusic often fail to sufficiently disentangle the target from the dense mixture, resulting in suboptimal performance on composite metrics. In contrast, Polyphonia achieves an optimal equilibrium, outperforming baselines on both ASB and AMB. This confirms that Polyphonia effectively enforces target semantics without compromising the structural fidelity quantified by LPAPS and CQT1-PCC.

This superiority is visually corroborated by the spectral comparison in Fig. 1 and **App. A**. More qualitative samples can be found at https://polyphonia2026.github.io/polyphonia-demo/.

## 5.3. Subjective Evaluation

Complementing the objective analysis, we conducted a Mean Opinion Score (MOS) study with 37 valid participants recruited from the MIR community (screened from $N = 50$). The test is approved by the Institutional ReviewBoard (IRB). Detailed screening protocols and interface designs are provided in **App. K**. As shown in Tab. 1, Polyphonia achieves the highest scores in TTA and GAC, confirming its ability to synthesize accurate targets within a coherent mix. Notably, while Melodia and SteerMusic yields a marginally higher CTI, this stems from its overly rigid constraints that prioritize preservation at the cost of editing effectiveness, as evidenced by their lower TTA. In contrast, Polyphonia demonstrates a superior trade-off of successful semantic transfer and background stability.

## 5.4. Ablation Study

**Module and Injection Layer Ablation**. Fig. 6 disentangles the contributions of our proposed modules and layer selections. We observe that removing Acoustic Modulation (w/o AM) results in high structural scores but poor target alignment, confirming that vanilla CA alone is insuf-

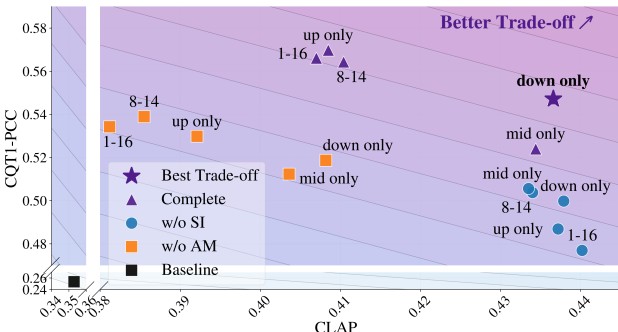

*Figure 6.* CLAP&CQT1-PCC trade-off analysis across different component configurations and layer selections. The Purple Star denotes our optimal setting (Complete model on all down layers), which occupies the upper-right Pareto frontier, indicating the best balance. w/o SI (Blue Circles) lacks Source Interpolation, sacrificing structure for alignment. w/o AM (Orange Squares) lacks Acoustic Modulation, failing to effectively inject target semantics.

ficient for precise targeting. Conversely, removing Source Interpolation (w/o SI) improves alignment at the cost of structural degradation. Our Complete framework applied to all down layers emerges as the optimal configuration, surpassing the 8-14 range used in Melodia (Yang et al., 2026). This aligns with findings in U-Net feature analysis (Frenkel et al., 2024), which suggest that down-sampling blocks encode high-level semantic layouts, while up-sampling blocks govern fine-grained textures. By selectively conditioning the down layers, we preserve the structural skeleton of the non-target stems while granting the up layers sufficient degrees of freedom to synthesize textural details required for the target timbre.

**Acoustic Prior Ablation**. Shown in Tab. 2, quantitative analysis is conducted to validate our acoustic prior mechanism from three perspectives: mask formulation, editing paradigm, and robustness to source separation quality.

*1) Mask Formulation (IRM vs. Norm):* Replacing IRM ($G_{X_0}$) with naive normalization ($G_{\text{norm}}$) leads to a performance degradation across all metrics. This confirms that probabilistic prior is essential for precise disentanglement.

*2) Editing Paradigm (Holistic vs. Sep-Remix):* We compare Polyphonia against "Separation-Editing-Remixing" baselines. While Sep-Remix strategies achieve lower LPAPS and KAD by mechanically preserving the non-target stems, their significantly lower CLAP scores reveal a failure to harmoniously blend the target semantics into the mixture. In contrast, Polyphonia achieves superior target alignment.

*3) Robustness to BSS Quality:* To assess whether our framework relies heavily on SOTA separation, we evaluate Polyphonia using lower-quality acoustic priors: **Unmix**, implemented via Open-Unmix (Stöter et al., 2019), and **Naive**, a DSP-based approach utilizing HPSS and frequency filtering

*Table 2.* Ablation study on types and qualities of acoustic priors and comparison with "separation-editing-remixing" (Sep-Remix) baselines. First , Second , and Third indicate the best, second best, and third best performance respectively.

| | Method | CLAP ↑ | CQT1-PCC ↑ | LPAPS ↓ | FAD ↓ | KAD ↓ | ASB ↑ | AMB ↑ |
|---|---|---|---|---|---|---|---|---|
| | SDEdit (Lower Bound) | 0.119±0.126 | 0.090±0.228 | 6.907±0.667 | 1.914±0.177 | 0.942±0.175 | 0.0000 | 0.0000 |
| **w/ IRM** | **Polyphonia+Demucs+$G_{X_0}$** | **0.437±0.101** | 0.547±0.225 | 4.096±0.643 | 0.949±0.233 | 0.695±0.238 | **0.8291** | **0.8631** |
| | Polyphonia+Unmix+$G_{X_0}$ | 0.435±0.099 | 0.541±0.228 | 4.173±0.637 | 0.953±0.231 | 0.696±0.240 | 0.8135 | 0.8543 |
| | Polyphonia+Naive+$G_{X_0}$ | 0.432±0.098 | 0.529±0.221 | 4.177±0.638 | 0.959±0.239 | 0.703±0.244 | 0.8097 | 0.8378 |
| **w/o IRM** | Polyphonia+Demucs+$G_{norm}$ | 0.413±0.105 | 0.459±0.229 | 4.194±0.650 | 1.229±0.241 | 0.728±0.242 | 0.7859 | 0.7372 |
| | Melodia (Sep-Remix) | 0.334±0.104 | **0.692±0.206** | **2.937±0.409** | **0.623±0.174** | **0.510±0.224** | 0.8068 | 0.8068 |
| | DDPM (Sep-Remix) | 0.330±0.117 | 0.532±0.217 | 3.887±0.683 | 0.869±0.182 | 0.648±0.228 | 0.7088 | 0.6971 |

(implementation details in **App. C.1**). Remarkably, Polyphonia maintains robust performance even with the Naive prior, significantly outperforming the "w/o IRM" baseline. This demonstrates that our Acoustic-Informed Attention Calibration is resilient to separation leakage and does not strictly require perfect source separation.

*4) Robustness across Stem Types:* To validate the effectiveness of our framework when dealing with non-standard stems, we evaluate the performance breakdown by stem types. As shown in Tab. 3, performance within the generic "Others" category is highly competitive with the primary "Vocals" category across both datasets. This demonstrates that even when utilizing a coarse "Others" mask as the acoustic prior, the cross-attention mechanism successfully resolves the target.

*Table 3.* Performance Breakdown by Stem Type

| Dataset | Stem Type | CLAP ↑ | LPAPS ↓ | CQT1-PCC ↑ |
|---|---|---|---|---|
| MUSDB18 | Vocals | 0.337 | 4.630 | 0.369 |
| | Others | 0.339 | 4.355 | 0.363 |
| MusicDelta | Vocals | 0.406 | 3.964 | 0.602 |
| | Others | 0.433 | 4.129 | 0.526 |

## 5.5. Additional Analysis

**Robustness to Prompt Sparsity** To evaluate Polyphonia's robustness for non-expert users, we constructed a *Sparse Prompt* set by stripping all detailed adjectives from the original *PolyEvalPrompts* (e.g., simplifying "A fast expressive violin..." to merely "A song with violin, bass, and drums"). As summarized in Tab. 4, while the target alignment exhibits an expected minor drop due to the lack of fine-grained textual guidance, Polyphonia with sparse prompts (CLAP: 0.322) still significantly outperforms state-of-the-art baselines even when they are equipped with full, descriptive prompts (e.g., Melodia's CLAP: 0.296).

**Performance under Varying Mixture Complexity** We further investigate the impact of mixture density on performance to identify potential bottlenecks in BSS-guided editing. As shown in Tab 5, Polyphonia maintains high stability across typical multi-track scenarios ($\leq 4$ tracks).

*Table 4.* Ablation study of prompt quality on MUSDB18-HQ. All baselines use full prompts, while Polyphonia is tested with both full and sparse prompts.

| Method | CLAP ↑ | LPAPS ↓ | CQT1-PCC ↑ | KAD ↓ | FAD ↓ |
|---|---|---|---|---|---|
| Melodia (Full) | 0.296 | 3.893 | 0.363 | 0.495 | 0.655 |
| SteerMusic (Full) | 0.255 | 4.105 | 0.383 | 0.497 | 0.747 |
| MusicGen (Full) | 0.295 | 6.600 | 0.003 | 0.840 | 1.374 |
| **Polyphonia (Full)** | 0.342 | 4.426 | 0.371 | 0.645 | 0.868 |
| **Polyphonia (Sparse)** | 0.322 | 4.372 | 0.363 | 0.629 | 0.863 |

*Table 5.* Ablation over varying number of stem tracks.

| Dataset | Track Count | CLAP ↑ | CQT1-PCC ↑ | LPAPS ↓ |
|---|---|---|---|---|
| MUSDB18-HQ | $\leq 3$ | 0.343 | 0.369 | 4.457 |
| | $= 4$ | 0.343 | 0.374 | 4.439 |
| | $\geq 5$ | 0.341 | 0.367 | 4.367 |
| MusicDelta | $\leq 3$ | 0.433 | 0.507 | 4.185 |
| | $= 4$ | 0.444 | 0.587 | 4.006 |
| | $\geq 5$ | 0.412 | 0.507 | 4.186 |

In extremely dense mixtures (e.g., MusicDelta samples with up to 9 tracks), we observe a slight performance decrease; however, the system exhibits *graceful degradation* rather than catastrophic failure.

## 6. Conclusion

We present Polyphonia, a framework resolving the semantic-acoustic misalignment through a coarse-to-fine calibration mechanism. By leveraging a probabilistic acoustic prior to establish coarse spectral boundaries, our approach guides the diffusion process to execute precise semantic synthesis, achieving state-of-the-art alignment and coherence. Crucially, Polyphonia unifies discriminative and generative paradigms by translating the hard outputs of source separation into soft inductive biases, thereby untangling the complex spectral interference that limits purely semantic approaches. While the current implementation inherits the inference latency of iterative diffusion and the generation ceilings of the pre-trained backbone, it establishes a flexible, model-agnostic paradigm. Future work will explore extending this decoupled logic to accelerated flow-matching architectures or open-vocabulary manipulation.

## Impact Statement

This work introduces Polyphonia, a framework that advances controllable music generation by enabling precise intra-stem editing; however, its application warrants careful consideration of potential societal and ethical implications. A primary concern stems from the reliance on pre-trained Blind Source Separation (BSS) models, which are predominantly trained on Western pop/rock taxonomies; this dependency may introduce inherent biases that marginalize non-Western musical traditions or instruments ill-defined by standard source categories. Furthermore, the capacity to surgically alter dense audio mixtures raises significant challenges regarding intellectual property and artist consent, as it facilitates the unauthorized modification or "remixing" of copyrighted material. Consequently, we advocate that the deployment of such powerful editing tools must go hand-in-hand with the development of robust provenance tracking and audio watermarking technologies to safeguard the integrity of artistic works and prevent potential misuse.

## Acknowledgment

This work was supported in part by the GJYC program of Guangzhou under Grant 2024D01J0081, in part by the ZJ program of Guangdong under Grant 2023QN10X455, and in part by the Fundamental Research Funds for the Central Universities under Grant 2025ZYGXZR053.

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

# A. Qualitative Comparison

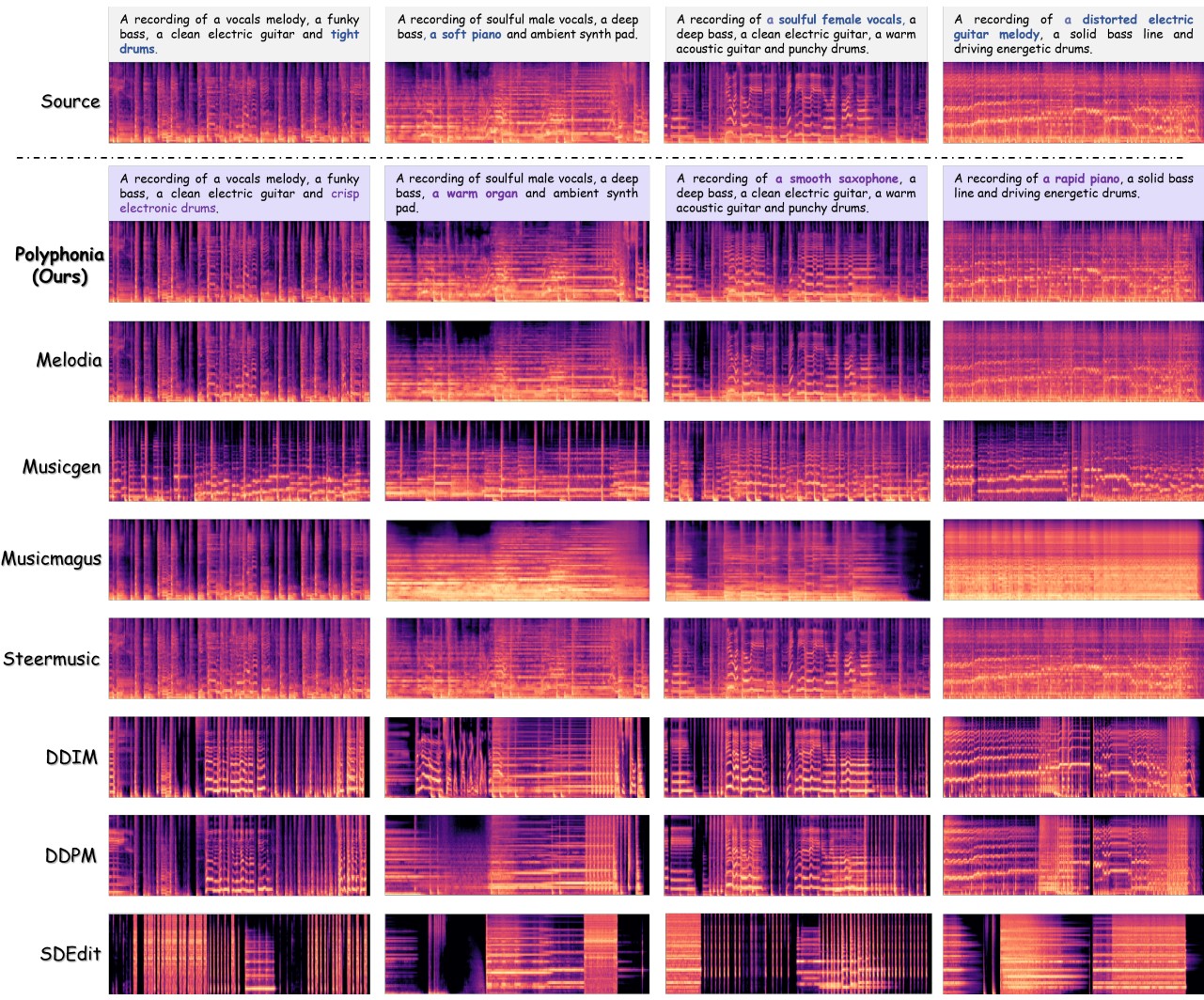

*Figure 7.* Qualitative Results of Polyphonia and baselines across different tasks.

## B. Pseudocode of *Polyphonia*

---

**Algorithm 1** Polyphonia: Zero-Shot Stem-Specific Timbre Transfer

---

**Require:** $X_0$: input music; $Y_{\text{tgt}}$: target prompt; $\lambda$: strength scalar; $\mathcal{L}$: selected layer set;

**Ensure:** $\hat{X}_0$: edited music

1: **1. Preparation Stage**
2: $z_0 \leftarrow \mathcal{E}(X_0)$
3: $G_{X_0} \leftarrow \text{EXTRACTIRM}(X_0, Y_{\text{tgt}})$ {Extract Acoustic Prior (Eq. 9)}
4: $\mathbf{m}^{\text{text}} \leftarrow \text{GETTOKENMASK}(Y_{\text{tgt}})$ {Binary mask for target tokens}
5: // Inversion must cache Logits $E$, not Maps $A$, for Pre-Softmax Interpolation
6: $z_T, \{E_{sa}^{\text{src}}, E_{loa}^{\text{src}}\} \leftarrow \text{DDIMINVERSION}(z_0)$
7: **2. Editing Stage**
8: **for** $t = T, \dots, 1$ **do**
9:     **Forward** T-UNet$(z_t, t, Y_{\text{tgt}})$ with **Attention Calibration**:
10:     **for** each Layer $l$ **do**
11:         $G \leftarrow \text{RESIZE}(G_{X_0}, l.\text{shape})$ {Align prior to layer resolution}
12:         **if** Layer $l \in \mathcal{L}$ **then**
13:             **for** each Attention Block **do**
14:                 *// Case 1: Text Cross-Attention → Acoustic Modulation (Eq. 13)*
15:                 **if** Block is Text-CA **then**
16:                     $\mathbf{B} \leftarrow \text{Flatten}(G) \otimes \mathbf{m}^{\text{text}}$ {Spatio-Textual Bias Matrix}
17:                     $E_{\text{text}} \leftarrow E_{\text{text}} + \lambda \cdot \mathbf{B}$ {Inject bias into Logits}
18:                     $A_{\text{text}} \leftarrow \text{SOFTMAX}(E_{\text{text}})$
19:                 **end if**
20:                 *// Case 2: SA & LoA Cross-Attention → Source Interpolation (Eq. 11)*
21:                 **if** Block is SA **or** LoA-CA **then**
22:                   $E_{\text{curr}} \leftarrow \frac{QK^{\top}}{\sqrt{d}}$
23:                     Retrieve cached source logits $E^{\text{src}}$ for this layer
24:                     // Pre-Softmax Injection (Raw Logit Space)
25:                     $E_{\text{mix}} \leftarrow (1 - G) \odot E^{\text{src}} + G \odot E_{\text{curr}}$
26:                     $A_{out} \leftarrow \text{SOFTMAX}(E_{\text{mix}})$
27:                 **end if**
28:             **end for**
29:         **end if**
30:     **end for**
31:     $z_{t-1} \leftarrow \text{DDIMSTEP}(z_t, \epsilon_\theta)$
32: **end for**
33: **return** $\mathcal{D}(z_0)$

---

## C. Additional Implementation Details

### C.1. Acoustic Prior Extraction and BSS Configurations

The calculation of the acoustic prior $G_{X_0}$ (Eq. 9) necessitates the decomposition of the input mixture $X_0$ into an estimated target stem $\tilde{S}_{\text{tgt}}$ and the remaining non-target stems $\tilde{S}_{\text{con}}$. To ensure acoustic consistency, the non-target stems is constructed in the waveform domain before spectral transformation. Let $y_{\text{mix}}$ denote the waveform of the input mixture. The separation model estimates the waveform stems $\{y_k\}_{k=1}^K$. Given a target instrument index $j$, the target waveform is $\tilde{y}_{\text{tgt}} = y_j$, and the non-target waveform is defined as the sum of all non-target components: $\tilde{y}_{\text{con}} = \sum_{k \neq j} y_k$. The final spectrograms $\tilde{S}_{\text{tgt}}$ and $\tilde{S}_{\text{con}}$ are then obtained via the Short-Time Fourier Transform (STFT): $\tilde{S} = |\text{STFT}(\tilde{y})|$.

Below, the specific configurations for the three source separation strategies utilized in the experiments and ablation studies are detailed.

**1) HT-Demucs (Default SOTA Prior).** For the main results, the Hybrid Transformer Demucs (**HT-Demucs**) (Rouard et al.,

2023) is employed. This state-of-the-art supervised model decomposes audio into four discrete waveform stems: `vocals`, `bass`, `drums`, and `others`. Target selection is performed by mapping the entity in the editing prompt to one of these categories (e.g., "violin" $\rightarrow$ `others`, "male voice" $\rightarrow$ `vocals`).

- **Standard Mapping**: If the target falls strictly into a primary category (e.g., `vocals`), then $\tilde{y}_{\text{tgt}} = y_{\text{vocals}}$ and $\tilde{y}_{\text{con}} = y_{\text{bass}} + y_{\text{drums}} + y_{\text{others}}$.

- **Hybrid Localization**: For instruments belonging to the generic `others` class, the Hybrid Localization Strategy is utilized (details in App. C.2).

**2) Open-Unmix (Mid-Tier Prior).** To assess robustness against lower-quality deep learning separation, **Open-Unmix (UMX)** (Stöter et al., 2019) is utilized. UMX is based on a bi-directional LSTM architecture and is pre-trained on the MUSDB18 dataset. Similar to Demucs, it outputs four waveform stems. However, compared to Transformer-based models, UMX typically exhibits higher spectral leakage and inter-source interference, serving as a representative baseline for mid-tier separation quality.

**3) Naive DSP (Low-Tier Prior).** To simulate a "worst-case" scenario where neural separation is unavailable or fails, a **Naive DSP** separator is implemented based on traditional signal processing heuristics. This method operates directly on the waveform or complex spectrum to produce time-domain outputs without learnable parameters:

- **Drums (Percussive)**: Extracted via Harmonic-Percussive Source Separation (HPSS) using a hard margin (margin=3.0). The percussive waveform is assigned to `drums`.

- **Bass (Low Harmonic)**: A $4^{\text{th}}$-order Butterworth low-pass filter with a cutoff frequency of 300 Hz is applied to the harmonic waveform component to isolate the `bass`.

- **Vocals/Others (Maximal Leakage)**: The remaining high-frequency harmonic content (residual waveform) inherently contains both vocals and melodic accompaniment. In this naive implementation, this identical residual mixture is assigned to both `vocals` and `others` stems (weighted by a scalar of 0.8 for `others`). This configuration creates a scenario of *maximal leakage*, where the acoustic prior $G_{\text{IRM}}$ derived from these waveforms acts as a coarse spectral envelope that encapsulates both the target vocals and the harmonic accompaniment, rather than isolating the target.

Consequently, this configuration tests the framework's ability to perform semantic selection within a permissive acoustic boundary. It demonstrates that even when the acoustic prior fails to fully disentangle the sources, it still successfully excludes structurally distinct components (e.g., drums and bass), allowing the attention mechanism to resolve the remaining ambiguity via semantic grounding.

### C.2. Hybrid Localization Strategy for Non-Standard Targets

A primary challenge in stem-based editing arises when the target instrument falls outside the standard taxonomy (Vocals, Bass, Drums, Others) utilized by the source separator. This is particularly common for instruments like piano or guitar, which, despite their functional similarity to vocals in terms of melodic importance, are typically aggregated into the "Others" stem. For transformations targeting instruments within the generic "Others" category (e.g., *guitar* $\rightarrow$ *violin* inside a mix with piano), we employ a **Hybrid Localization Strategy**. Specifically, we designate the entire "Others" stem as the target reference $\tilde{S}_{\text{tgt}}$. The derived acoustic prior $G_{\text{IRM}}$ thus acts as a coarse but safe permissive constraint, effectively filtering out the non-target energy of structurally distinct components like Bass and Drums. While pure vanilla attention fails to isolate targets in the global dense mixture due to severe spectral interference, **cross-attention functions effectively within the reduced search space provided by the IRM**. Inside this constrained region, the spectral interference is significantly mitigated, allowing the semantic selectivity of the cross-attention mechanism to resolve remaining local ambiguities. For instance, when prompting for "Violin", the attention heads naturally attend to the harmonic structures of the guitar (within the "Others" region) while ignoring the percussive decay of the piano, achieving precise disentanglement without suffering from target isolation failure.

### C.3. Token Mask Construction

To construct the binary token mask $\mathbf{m}^{\text{text}}$, we align the masking logic directly with the T5 tokenizer used by the AudioLDM 2 text encoder. For multi-token target words (e.g., "upright bass" tokenized as `["up", "right", "bass"]`), we identify

the continuous span of indices corresponding to the target phrase within the tokenized sequence and assign a value of 1 to all associated positions in $\mathbf{m}^{\text{text}}$. We strictly map the mask to the tokens present in the user's input prompt without employing external synonym dictionaries or latent expansion. This direct mapping ensures that the acoustic modulation boosts the precise semantic concepts specified by the user, avoiding potential drift caused by vocabulary mismatches between the tokenizer and external knowledge bases.

### C.4. Tensor Alignment and Broadcasting Logic

The integration of the 2D acoustic prior into the 4D attention tensors is implemented via specific tensor broadcasting mechanisms tailored to the attention type.

**Case 1: Text Cross-Attention (Acoustic Modulation).** As formulated in Eq. 13, we compute a Spatio-Textual Bias Matrix $\mathbf{B} = \mathbf{g} \otimes \mathbf{m}^{\text{text}} \in \mathbb{R}^{L_z \times L_y}$. Since the standard Multi-Head Attention score matrix $E$ possesses the shape $(B \cdot H, L_z, L_y)$, the bias matrix $\mathbf{B}$ is unsqueezed to dimensions $(1, 1, L_z, L_y)$ and broadcasted across the batch and head dimensions. This implementation ensures that all attention heads are globally encouraged to focus on the target acoustic region defined by $G$, preventing individual heads from diverging to background regions due to semantic ambiguity.

**Case 2: Self and LoA Attention (Source Interpolation).** For Source Interpolation (Eq. 10), the attention energy matrix resides in $\mathbb{R}^{L_z \times L_z}$. We implement a **row-wise per query** broadcasting mechanism. Let $g_i$ be the $i$-th element of the flattened acoustic prior $\mathbf{g} \in \mathbb{R}^{L_z}$. The mixed energy for query $i$ and key $j$ is computed as:

$$(E_{\text{mix}})_{i,j} = (1 - g_i)(E_{\text{src}})_{i,j} + g_i(E_{\text{curr}})_{i,j} \tag{15}$$

The selection of the row-wise broadcasting dimension is a strict requirement for Polyphonia's structural integrity. This ensures that the decision to preserve or edit is governed by the *query's* spatial-spectral location (i.e., where the model is currently synthesizing features).

### C.5. Hyperparameter and Scheduling Configuration

Regarding the temporal and architectural application of the proposed mechanisms, we adopt a static configuration to ensure zero-shot stability. The modulation scalar is set to $\lambda = 2.5$ and remains constant across all inference timesteps. As shown in Tab. 6, while time-step decay strategies (Linear or Cosine) marginally improve structural preservation (LPAPS, CQT1-PCC), a constant schedule maximizes target alignment (CLAP).

*Table 6.* Ablation on $\lambda$ Time-Step Scheduling (max $\lambda = 2.5$)

| Schedule Type | CLAP ↑ | LPAPS ↓ | CQT1-PCC ↑ | KAD ↓ | FAD ↓ |
|---|---|---|---|---|---|
| Constant | **0.437** | 4.096 | 0.547 | 0.695 | 0.949 |
| Linear Decay | 0.435 | 4.083 | **0.549** | **0.686** | 0.942 |
| Cosine Decay | 0.435 | **4.081** | 0.549 | 0.686 | **0.941** |

Furthermore, we maintain a uniform $\lambda$ across all down-sampling layers. The theoretical rationale is that the acoustic prior $G$ represents an objective physical energy boundary. Varying the modulation strength $\lambda$ per layer would mathematically imply that the physical location of the stem shifts at different abstraction levels, which contradicts the consistent spatial-spectral nature of audio sources.

## D. Efficiency Analysis

To evaluate the computational cost of our proposed framework and demonstrate its scalability, we benchmarked the inference time and memory consumption of Polyphonia against all baseline methods across different audio lengths (5s, 10s, and 15s). The measurements were conducted on a single NVIDIA GeForce RTX 3090 (24GB). We report the breakdown of time costs into Pre-processing (BSS and Mask generation), Inversion, and Generation stages.

**Inference Speed Analysis.** As shown in Table 7, Polyphonia exhibits consistent computational efficiency that scales well with audio length. For a standard 10-second generation, Polyphonia achieves a total inference time of 24.38 seconds. While this represents a marginal increase compared to *Melodia* (20.01s), this overhead is primarily attributed to the one-time Acoustic Prior Extraction via Demucs (1.52s) and the additional matrix operations required for Attention Calibration.

*Table 7.* **Efficiency Comparison across different audio lengths.** Runtime and peak VRAM usage are measured for generating 5s, 10s, and 15s audio samples (100 diffusion steps). Polyphonia demonstrates robust performance scaling without drastic memory overhead.

| Method | Audio Length | Pre-process (s) | Inversion (s) | Generation (s) | Total Time (s) | Peak VRAM (GB) |
|---|---|---|---|---|---|---|
| SDEdit | 5s | - | 0.48 | 7.86 | 8.34 | 4.39 |
| | 10s | - | 0.50 | 9.50 | 10.00 | 4.49 |
| | 15s | - | 0.56 | 12.57 | 13.13 | 4.60 |
| MusicGen | 5s | - | - | 8.89 | 8.89 | 4.19 |
| | 10s | - | - | 17.93 | 17.93 | 4.21 |
| | 15s | - | - | 23.67 | 23.67 | 4.28 |
| DDIM | 5s | - | 9.32 | 9.43 | 18.75 | 4.41 |
| | 10s | - | 9.81 | 9.35 | 19.16 | 4.50 |
| | 15s | - | 10.87 | 10.74 | 21.61 | 4.57 |
| DDPM | 5s | - | 9.79 | 9.61 | 19.40 | 4.42 |
| | 10s | - | 10.01 | 9.59 | 19.60 | 4.52 |
| | 15s | - | 10.38 | 10.18 | 20.56 | 4.62 |
| Melodia | 5s | - | 9.89 | 9.62 | 19.51 | 6.92 |
| | 10s | - | 10.13 | 9.88 | 20.01 | 9.62 |
| | 15s | - | 11.35 | 10.82 | 22.17 | 12.93 |
| MusicMagus | 5s | - | 43.56 | 34.88 | 78.44 | 6.95 |
| | 10s | - | 44.33 | 36.09 | 80.42 | 8.69 |
| | 15s | - | 49.75 | 46.16 | 95.91 | 11.47 |
| SteerMusic | 5s | - | 4.26 | 48.48 | 52.74 | 4.51 |
| | 10s | - | 4.26 | 51.97 | 56.23 | 4.64 |
| | 15s | - | 4.26 | 73.12 | 78.38 | 4.77 |
| **Polyphonia (Ours)** | 5s | 1.27 | 10.79 | 10.54 | 22.60 | 5.30 |
| | 10s | 1.52 | 11.53 | 11.33 | 24.38 | 8.05 |
| | 15s | 1.61 | 10.88 | 12.02 | 23.90 | 13.68 |

Crucially, this slight increase in latency yields a significant performance leap, resolving the non-target distortion issues that plague faster, unguided methods like DDPM and SDEdit. Furthermore, compared to optimization-based editing paradigms such as *SteerMusic* (56.23s) and *MusicMagus* (80.42s) at 10 seconds, Polyphonia is approximately **2.3× to 3.3× faster**. This confirms that our feed-forward attention calibration mechanism is significantly more efficient than iterative gradient-based guidance across varying context lengths while delivering superior stem-specific timbre transfer results.

**Memory Consumption Analysis.** Regarding memory usage, we must first highlight a critical correction from earlier manuscript versions: the peak VRAM for Polyphonia was initially artificially inflated due to a code bug that redundantly stored unused tensors during the calibration stage. We have explicitly resolved this memory leak, a fix which significantly reduces the peak footprint **without affecting any generation results or inference time**.

As reflected in the updated Table 7, Polyphonia now accurately requires roughly **8.05 GB** of peak VRAM for a 10-second clip. While the framework necessitates holding the Blind Source Separation (BSS) model in memory alongside the diffusion backbone, this corrected footprint remains highly accessible for standard consumer-grade GPUs (e.g., RTX 3090/4090). Moreover, as audio length scales up to 15 seconds, Polyphonia handles the expansion gracefully with a peak VRAM of 13.68 GB. The results demonstrate that injecting the BSS acoustic prior achieves superior and strictly localized editing performance without drastically increasing computation time or memory requirements.

# E. Hyperparameter Sensitivity Analysis

We perform a grid search to identify the optimal Acoustic Modulation scalar $\lambda$ (Eq. 13) and Classifier-Free Guidance (CFG) scale, aiming to maximize target alignment (CLAP) while minimizing structural deviation (LPAPS) and preserving musicality (CQT1-PCC).

Regarding the acoustic modulation scalar $\lambda$, Figure 8a illustrates that increasing $\lambda$ up to 2.5 yields steady improvements in target alignment (CLAP) while maintaining stable structural fidelity (CQT1-PCC). However, surpassing this threshold causes a sharp collapse in structural integrity and a disproportionate increase in perceptual distance (LPAPS) with negligible semantic gains, leading us to select $\lambda = 2.5$ as the optimal configuration.

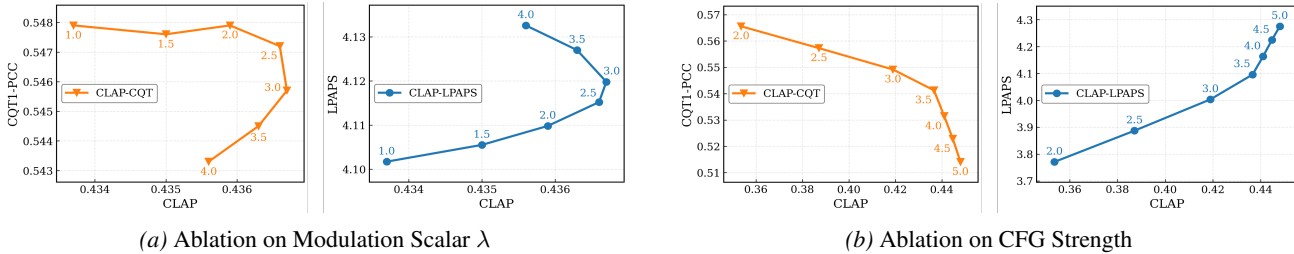

(a) Ablation on Modulation Scalar $\lambda$        (b) Ablation on CFG Strength

*Figure 8.* Hyperparameter sensitivity analysis. The curves illustrate the trade-off between target alignment (CLAP, x-axis) and structural integrity (CQT/LPAPS, y-axis). We select the values that represent the "elbow points" of these curves.

In terms of CFG strength, Figure 8b reveals a distinct trade-off where lower values favor structural retention but result in weak editing effects. The curve exhibits a prominent "elbow" at CFG $= 3.5$; pushing guidance beyond this point yields diminishing returns for target alignment while causing rapid deterioration in musicality, indicating that the generation begins to disregard the source audio constraints. Consequently, we fix CFG $= 3.5$ for all experiments.

## F. Details of Shannon Entropy Analysis

To quantitatively evaluate the concentration ("sharpness") of the attention mechanisms, we utilize the **Shannon Entropy** (Shannon, 1948) as a metric of uncertainty. In the context of attention maps, a lower entropy value indicates a more peaked distribution, signifying that the model is confidently attending to specific structural features. Conversely, higher entropy implies a flatter, more diffuse distribution.

Formally, let $A^{(l,h)} \in \mathbb{R}^{N \times M}$ denote the attention probability matrix for the $h$-th head in layer $l \in \mathcal{L}$, where $\mathcal{L}$ is the set of analyzed layers, $N_h$ is the number of heads, $N$ is the number of queries, and $M$ is the number of keys. The entropy for the $i$-th query row is calculated as:

$$H(A_i^{(l,h)}) = - \sum_{j=1}^{M} A_{i,j}^{(l,h)} \log(A_{i,j}^{(l,h)} + \epsilon) \tag{16}$$

where $\epsilon = 10^{-12}$ is a small constant for numerical stability. The final reported entropy for a given timestep is the average over all queries, heads, and targeted layers:

$$\text{Entropy} = \frac{1}{|\mathcal{L}| \cdot N_h \cdot N} \sum_{l \in \mathcal{L}} \sum_{h=1}^{N_h} \sum_{i=1}^{N} H(A_i^{(l,h)}) \tag{17}$$

We compute this metric across diffusion steps to visualize the dynamics of attention across diffusion time-steps in Fig. 5.

## G. Baseline Implementation Details

To ensure a rigorous and fair comparison, all baseline methods are implemented using their official open-source repositories or reproducible scripts. **Note on Inference Steps:** While our proposed *Polyphonia* achieves optimal performance with only **100 diffusion steps**, we execute all diffusion-based baselines using **200 steps** (and optimization-based methods up to 400 steps) to ensure they converge to their highest quality, following the standard configurations reported in their respective literature (Yang et al., 2026; Manor & Michaeli, 2024).

All diffusion-based methods utilize the pre-trained **AudioLDM 2** checkpoint (`cvssp/audioldm`, 1.1B parameters) as the backbone. The specific configurations for each baseline are detailed below:

**MusicGen (Copet et al., 2023).** As the autoregressive upper bound, we utilize the `facebook/musicgen-melody` checkpoint (Medium, 1.5B parameters). Inference is performed with a guidance scale of 3.0 and a duration of 10 seconds. We condition the generation on the chromagram extracted from the source mixture to preserve melodic structure.

**Melodia (Yang et al., 2026).** We implemented the method from scratch to mirror the official design choices. The inference runs for 200 steps with a full inversion start point ($T_{\text{start}} = 1000$). The Target Classifier-Free Guidance (CFG) scale is set to 5.5, consistent with the optimal balance point reported by the authors. Crucially, the Self-Attention (SA) injection is applied specifically to Layers 8-14 of the T-UNet.

**SteerMusic (Niu et al., 2026).** Based on the official implementation script, this method employs an optimization-based guidance approach. We set the validation steps (total optimization loop) to **400 steps** using an SGD optimizer with a learning rate of **0.02**. The energy guidance scale is set to $\lambda_{\text{energy}} = 30$, and the augmentation weight is $w_{\text{aug}} = 2$. The guidance is applied within the time range of $[0.02, 0.98]$.

**MusicMagus (Zhang et al., 2024).** We adopt the official inference pipeline combining DDIM inversion with optimization. Following comparative protocols (Yang et al., 2026), we perform editing with $T_{\text{start}} = 1000$ (200 steps). The Target CFG is set to **3.5**, and the Source CFG is set to **1.0**.

**DDIM Inversion (Song et al.).** This serves as the deterministic reconstruction baseline. We perform a full inversion ($T_{\text{start}} = 1000$) over 200 steps. During the denoising generation, we use a Target CFG of **5.0** and a Source CFG of **3.0** to balance text adherence and reconstruction.

**DDPM-Friendly (Manor & Michaeli, 2024).** We employ the stochastic inversion schedule tailored for audio. Following the recommendations in (Yang et al., 2026) and (Manor & Michaeli, 2024), we set the partial starting timestep to $T_{\text{start}} = 500$ (i.e., skipping the first 50% of steps) to preserve structure. The Target CFG is set to **12.0**, and the Source CFG is set to **3.0**.

**SDEdit (Meng et al.).** We apply the standard stochastic encoding-decoding strategy. Similar to DDPM-Friendly, we corrupt the source audio with Gaussian noise up to $t = 500$ ($T_{\text{start}} = 500$) and denoise for 200 steps. The Target CFG is set to **12.0** and Source CFG to **3.0** to maximize editability.

**Note on PPAE Exclusion.** Although PPAE (Xu et al., 2024) is a contemporary zero-shot editing method, we exclude it from the quantitative benchmark for two primary reasons. First, there is no official open-source code or public hyperparameter configuration available to ensure an accurate and fair reproduction on multi-track music datasets. Second, in our internal pilot reproduction attempts, we observed significant *Non-Target Distortion*, where non-target stems were severely distorted and the original melodic structure was compromised. To maintain the rigor of our comparative analysis, we only include baselines with verifiable and stable implementations.

## H. Detailed Dataset Statistics

We provide the distribution of track counts for both evaluation benchmarks to contextualize the complexity of the stem-specific timbre transfer tasks.

*Table 8.* Song Type Distribution by Stem Count in MUSDB18-HQ and MusicDelta

*(a)* MUSDB18-HQ (99 unique stems)

| Song Type | Count |
|---|---|
| 1-stem | 1 |
| 2-stem | 1 |
| 3-stem | 13 |
| 4-stem | 23 |
| 5-stem | 10 |
| 6-stem | 2 |
| **Total** | **50** |

*(b)* MusicDelta (46 unique stems)

| Song Type | Count |
|---|---|
| 3-stem | 12 |
| 4-stem | 14 |
| 5-stem | 1 |
| 9-stem | 1 |
| **Total** | **28** |

## I. Additional Details on *PolyEvalPrompts*

### I.1. Pipeline and Prompts

Our prompt set generation pipeline operates in two sequential stages to ensure both acoustic accuracy and task diversity. We utilize the Qwen family of models for both stages.

**Stage 1: Acoustic Analysis (Audio-to-Text).** In this stage, we employ `Qwen-Audio` (specifically `qwen3-omni-flash`) (Chu et al., 2023) to analyze the raw waveforms. Unlike text-only models, Qwen-Audio can directly "listen" to the stems and extract fine-grained acoustic details. The exact system prompt used in our script (`preprocess_musdb18.py`) is shown in Figure 9.

```
System Instruction:
Listen to this audio stem. Describe all the instruments used, the playing technique
used (e.g., picked, fingered, aggressive, soft), the tempo (fast/slow), and the genre.
Output in JSON format.
─────────────────────────────────────────────────────────────────────────────
User Input:
Analyze this audio. [Audio Embedding]
```

*Figure 9.* Stage 1 Prompt: Querying Qwen-Audio for acoustic metadata extraction.

**Stage 2: Task Synthesis (Text-to-Text).** Conditioned on the metadata extracted in Stage 1, we use `Qwen-Plus` (Yang et al., 2025) to procedurally generate the editing tasks. We designed a strict protocol to enforce format standardization and logical consistency (e.g., the "Mirror Rule" for softmasks). The exact system prompt used in our script (`preprocess_text_musdb18.py`) is shown in Figure 10.

```
System Instruction:
You are a Strict Audio Dataset Engineer. Generate a 'prompt_multi.json' with exactly 15 tasks.

### PROTOCOL 1: SIMPLIFICATION & SANITIZATION
1. **ONE ADJECTIVE ONLY**: Use exactly ONE adjective per instrument.
     - Bad:  "a synth bass with aggressive and distorted tone"
     - Good: "a gritty synth bass"
2. **NO FORBIDDEN CHARS**: Replace "/" with " and ". No "(", ")", "[", "]".

### PROTOCOL 2: SOURCE ANALYSIS
Define the 4 stems based on metadata:
- **Vocals**: If instrumental, set to empty string '""'. Else, e.g., "soulful male vocals".
- **Others**: Main melodic instrument (e.g., "distorted guitar").
- **Bass**: e.g., "deep bass".
- **Drums**: e.g., "punchy drums".

### PROTOCOL 3: TASK ALLOCATION
**Scenario A (Has Vocals):**
- 7 Vocals Tasks (e.g., 'vocals2violin')
- 5 Others Tasks (e.g., 'guitar2piano')
- 2 Drums Tasks (e.g., 'drums2electronic')
- 1 Bass Task (e.g., 'bass2upright')

**Scenario B (Instrumental / No Vocals):**
- 0 Vocals Tasks
- 9 Others Tasks (HEAVILY modify the main instrument, e.g., 'guitar2flute', 'guitar2choir')
- 4 Drums Tasks
- 2 Bass Tasks

### PROTOCOL 4: GENERATION LOGIC (The "Mirror" Rule)
For each task:
1. **Target**: Define the new description (e.g., "a smooth violin").
2. **Original Prompt**: "A recording of {Vocals}, {Bass}, {Others} and {Drums}."
3. **Baseline Prompt**: Replace the target track in Original Prompt with the NEW description.
4. **Softmasks**: Copy the EXACT phrases from the **Baseline Prompt**.
     - *Crucial*: If you edit Others to "a smooth violin", the 'Others Softmask Prompt' MUST be "a smooth violin".
     - If a track is empty (like Vocals in instrumental), keep it '""'.

### EXAMPLE (Instrumental Case)
"guitar2violin": {
    "Original Prompt": "A recording of a deep bass, a distorted guitar and punchy drums.",
    "Baseline Prompt": "A recording of a deep bass, a smooth violin and punchy drums.",
    "Vocals Softmask Prompt": "",
    "Bass Softmask Prompt": "a deep bass",
    "Drums Softmask Prompt": "punchy drums",
    "Others Softmask Prompt": "a smooth violin"
}
─────────────────────────────────────────────────────────────────────────────
User Input:
Metadata: {JSON Metadata Object}
```

*Figure 10.* Stage 2 Prompt: Querying Qwen-Plus to generate structured editing tasks based on the extracted metadata.

### I.2. Prompt Examples

We present specific examples of the generated *PolyEvalPrompts* benchmark entries in Tab. 9.

*Table 9.* Examples of audio-prompt pairs from MusicDelta Dataset

| Music Sample | Prompts |
| --- | --- |
| **[Zeppelin]** | **Source Prompt:** A recording of a distorted electric guitar melody, a solid bass line and driving energetic drums. 
 **Edit Prompt 1 (Guitar → Violin):** A recording of a fast expressive violin, a solid bass line and driving energetic drums. 
 **Edit Prompt 2 (Guitar → Cello):** A recording of a powerful cello, a solid bass line and driving energetic drums. 
 ... |
| **[Vivaldi]** | **Source Prompt:** A recording of a deep double bass line, a flowing cello melody, a smooth viola harmony and a lively violin lead. 
 **Edit Prompt 1 (Bass → Piano):** A recording of a resonant piano, a flowing cello melody, a smooth viola harmony and a lively violin lead. 
 **Edit Prompt 2 (Bass → Synth):** A recording of a warm analog synth, a flowing cello melody, a smooth viola harmony and a lively violin lead. 
 ... |
| **[Disco]** | **Source Prompt:** A recording of smooth vocals, a steady bass line, a clean rhythmic electric guitar melody and a tight, driving drum beat. 
 **Edit Prompt 1 (Vocals → Violin):** A recording of a violin melody, a steady bass line, a clean rhythmic electric guitar melody and a tight, driving drum beat. 
 **Edit Prompt 2 (Vocals → Trumpet):** A recording of a trumpet melody, a steady bass line, a clean rhythmic electric guitar melody and a tight, driving drum beat. 
 ... |
| **...** | ... |

## J. Implementation Details of Objective Metrics.

To ensure a rigorous and reproducible evaluation, we adopt standard objective metrics following the implementation details established in recent state-of-the-art works (Yang et al., 2026; Niu et al., 2026).

**Text-Audio Alignment (CLAP Score).** We evaluate the semantic alignment between the edited audio and the target text prompt using the CLAP metric. Following previous methods, we utilize the official LAION-CLAP model with the checkpoint `music_audioset_epoch_15_esc_90.14.pt` (Wu et al., 2023). Since this model was trained on 10-second audio segments, assessing variable-length or long-form music requires a windowing strategy. We adopt the sliding window approach where the input audio is split into 10-second segments with a 1-second overlap. The final score for a track is calculated as the average score of all segments. Higher CLAP values indicate stronger semantic alignment.

**Structural Consistency (LPAPS).** To quantify the preservation of temporal structure and perceptual coherence between the source and edited audio, we employ the Learned Perceptual Audio Patch Similarity (LPAPS). Unlike standard implementations that use VGG-based features, we follow the music-specific adaptation which utilizes intermediate features from the Swin-Transformer blocks of the pre-trained CLAP model (specifically the same checkpoint used for CLAP Score) (Manor & Michaeli, 2024; Paissan et al., 2024). Lower LPAPS values indicate better structural preservation.

**Melodic Consistency (CQT1-PCC).** To explicitly measure melodic preservation beyond general structural similarity, we utilize the Top-1 Constant-Q Transform Pearson Correlation Coefficient (CQT1-PCC). We extract CQT features using the `nnAudio` library with the `CQT2010` function. The parameters are set to a sampling rate of 16 kHz, `n_bins=128`, and `bins_per_octave=24`. We extract the top-1 frequency bin, which contains the most dominant melodic information, and calculate the Pearson correlation coefficient between the source and edited audio. Higher values indicate stronger melodic fidelity.

**Distributional Similarity (FAD and KAD).** To comprehensively evaluate the distributional alignment between the generated audio and the reference music, we employ both Fréchet Audio Distance (FAD) and Kernel Audio Distance (KAD). FAD measures the distance between the reference and generated distributions by approximating them as multivariate Gaussians in a high-dimensional embedding space. KAD, on the other hand, computes the Maximum Mean Discrepancy (MMD) between the feature distributions using polynomial kernels, providing an unbiased estimator of the distance. Crucially, for both metrics, we utilize the **LAION-CLAP** model as the feature extractor to ensure alignment with the semantic evaluation space. Specifically, we use the same checkpoint `music_audioset_epoch_15_esc_90.14.pt` as used in the CLAP Score calculation (Yang et al., 2026; Niu et al., 2026). For the calculation of FAD, we follow the implementation provided in the `fadtk` toolkit (Niu et al., 2026). Lower FAD and KAD values indicate closer distributional alignment.

## K. Subjective Evaluation Details

Stem-specific timbre transfer presents unique perceptual challenges that standard audio quality metrics fail to capture. To address this, our subjective study was conducted under a rigorous protocol designed to evaluate specific success criteria—target fidelity, non-target integrity, and overall coherence—rather than general audio quality.

### K.1. Metric Definitions

We designed the questionnaire to map specific perceptual dimensions to the editing task. Participants rated samples on a 5-point Likert scale (1: Bad to 5: Excellent) based on the following defined dimensions.

**Target Timbre Alignment (TTA).** This metric measures the fidelity of the target stem's transformation. In the evaluation interface, participants were explicitly asked: *"Does the instrument or timbre in this edited music actually sound like what the target description asks for?"* Raters assessed whether the specific instrument specified in the prompt (e.g., "violin") had been successfully synthesized and was clearly identifiable, distinguishable from the source timbre.

**Contextual Timbre Integrity (CTI).** To address the "leakage" failure mode common in mixture editing, CTI strictly evaluates the preservation of non-target background tracks. The corresponding question presented to raters was: *"Besides the main sound we're trying to change, does the background music (accompaniment, drums, etc.) stay the same?"* A high score indicates that the accompaniment remains perceptually unchanged and free from artifacts or inadvertent morphing (e.g., drums taking on tonal characteristics of the target).

**Global Acoustic Coherence (GAC).** This metric assesses the overall harmonic and spectral blend between the newly generated target stem and the preserved non-target stems. It specifically penalizes "cut-and-paste" artifacts often found in separation-remixing baselines. Participants answered the question: *"Does the edited sound blend naturally with the background music?"* to ensure the result sounds like a cohesive musical piece rather than disjointed stems.

### K.2. Experimental Procedure

To ensure the statistical validity and professional rigor of our subjective evaluation, we implemented a strict screening and testing protocol approved by the Institutional Review Board (IRB).

**Participant Screening.** We recruited an initial pool of 50 participants from the MIR community and professional music forums. To ensure data quality, we applied comprehensive screening criteria. As shown in Figure 11, we collected demographic data including age, gender, and musical background. We required participants to categorize their musical experience into three levels: Amateur, Intermediate, or Professional. Only participants with at least "Amateur" experience—defined as having basic music theory knowledge and listening experience—were qualified.

**Data Filtering.** To maintain the integrity of the results, we embedded "Attention Checks" (obvious ground-truth samples) within the test. Participants who failed these checks or completed the survey in unrealistic timeframes (significantly deviating from the estimated 25-minute session duration) were excluded. After this rigorous filtering, 37 valid response sets were retained for analysis. Furthermore, to prevent loudness bias during the listening phase, all audio samples—including source recordings and outputs from all baseline models—were loudness-normalized to -19.11 dBFS (LUFS). The samples were presented in a randomized order to eliminate sequential bias.

### K.3. Evaluation Interface

We developed a customized web-based evaluation interface to streamline the listening test. The workflow proceeded in three stages. First, participants completed a survey regarding their musical background and study duration (Figure 11). Second, they reviewed a detailed instruction guide (Figure 12) defining the concepts of "Source Music" and "Target Prompt" to ensure they understood the stem-specific timbre transfer objective. Finally, in the listening and rating phase, participants evaluated a total of 15 distinct musical samples. For each sample, raters compared the outputs of 8 different methods (Polyphonia and 7 baselines) alongside the source reference. Both source and target textual prompts were displayed to clarify the editing intention.

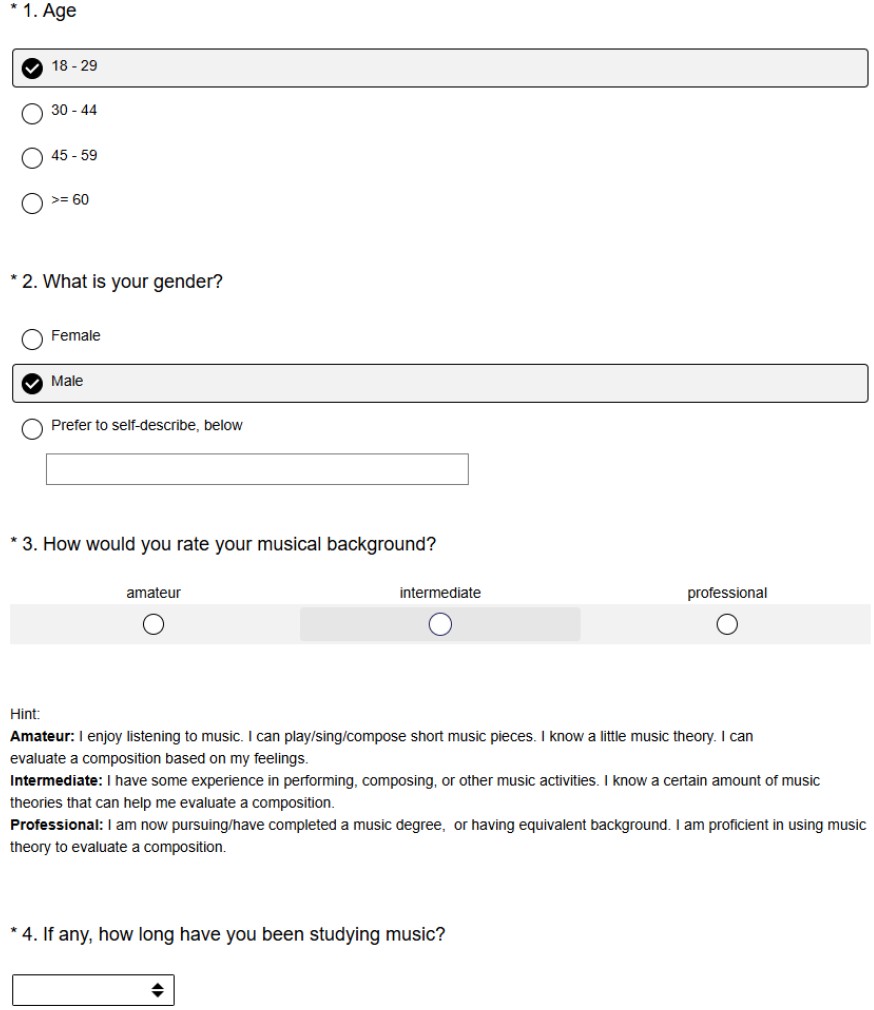

*Figure 11.* Screenshot of the demographic and musical background screening interface. Participants were required to self-assess their musical proficiency.

## Instructions

Welcome! In this short listening test, you'll help us evaluate some edited music clips. Your evaluation will be of great help to us.

**What you'll do:**

- **First, you'll hear the "Source Music"** — this is the original music clip that we want to modify.

- **You'll see a "Source Prompt"** — a text description of what the original music sounds like (e.g., "A recording of soulful male vocals").

- **You'll see a "Target Prompt"** — a description of how we want to change the music (e.g., "A recording of bright flute").

- **Then, you'll hear an "Edited Result"** — this is the computer's attempt to apply that change. Your task is to judge how well it worked.

**What to expect:** You may hear very similar audio clips across different results. This is normal and expected. Please trust your intuition and rate each result based on what you hear.

**Please use headphones** for the best listening experience. There are no right or wrong answers—we value your honest opinion. Thank you for your help!

*Figure 12.* Screenshot of the instruction page, defining the task terminology and ensuring participants understood the stem-specific timbre transfer objective.

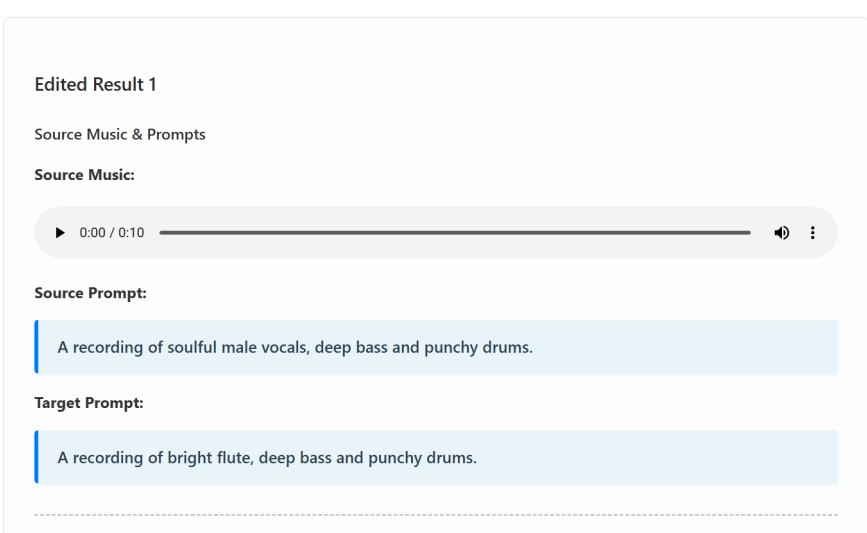

*Figure 13.* Screenshot of the listening test interface. The source audio and prompts serve as the reference standard for evaluating the edited result.

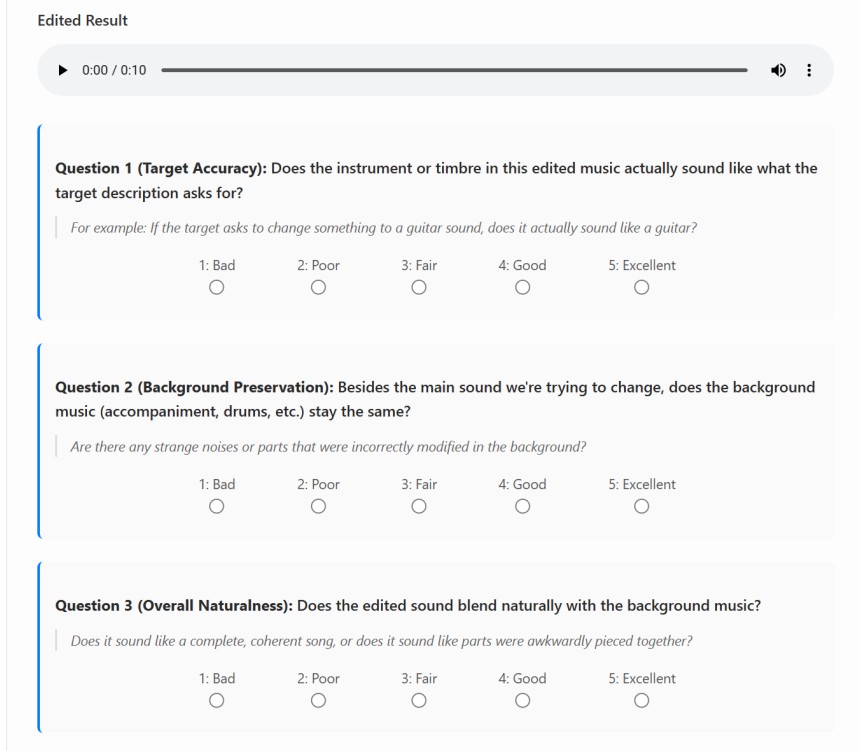

*Figure 14.* Screenshot of the evaluation questionnaire. The three questions displayed correspond directly to our TTA, CTI, and GAC metrics.

