# OpenReview forum: "Polyphonia: Zero-Shot Timbre Transfer in Polyphonic Music with Acoustic-Informed Attention Calibration"
_ICML.cc/2026/Conference — ICML 2026 regular_

### Official Review · Reviewer_hMfC · 2026-03-05

**Soundness:** 2
**Presentation:** 3
**Significance:** 3
**Originality:** 2
**Overall Recommendation:** 4
**Confidence:** 4

**Summary:**

This paper presents Polyphonia, a system for music editing in the context of source-specific timbre transfer in polyphonic audio mixtures. This relies on an acoustic prior via an ideal ration mask obtained through source separation, and an acoustic informed attention calibration. Experiments are carried out on the MUSDB18 and MusicDelta datasets, and comparisons are made with other music editing models covering both objective and subjective metrics.

**Compliance With Llm Reviewing Policy:**

Affirmed.

**Final Justification:**

Overall, this is a well written paper which has a good motivation and justification. The evaluation is overall solid. In my review I had comments on the terminology used which to a large extent have been addressed. Other comments on clarity and the experimental procedure have to a large extent been addressed. My overall recommendation has now moved towards weak acceptance.

**Key Questions For Authors:**

No specific questions to ask the authors.

**Limitations:**

yes

**Strengths And Weaknesses:**

Strengths:
* Well written and clear paper, citing and discussing relevant work.
* Technically appropriate methodology, with good justification and motivation.
* Thorough experiments on well known open datasets using both subjective and objective metrics, and good comparison with other state-of-the-art methods.

Weaknesses:
* The terminology used in this paper is sometimes confusing, and does not correspond with terminology used in other papers covering similar topics. For example terms such as "context locking", "target ineffectiveness", "context-aware intra-stem editing", "external acoustic anchoring", "semantic intent" are non-standard and sometimes refer to existing issues and tasks already described in the literature using different terms. To provide a concrete example, the "context-aware music editing" task referred to here is essentially timbre transfer in polyphonic audio. Terminology needs to be thoroughly revised as for the paper to be in line with others in the literature and as for the paper to be as accessible as possible to the reader.
* The fact that source separation is one of the main components of the proposed system is not properly specified until this is first used in section 4.1.2. This would need to be signposted earlier in the paper starting from the abstract and introduction since this is a crucial system component.
* Section 4.1.2: it is unclear why a simple baseline with "separation-editing-remixing" would necessarily lack coherence. If the editing process takes care to preserve the pitch of the original source instrument to be edited, then I would not anticipate any major issues with coherence. Unless the editing process can radically alter the melody to be transferred, in which case this needs to be explicitly stated in the paper. In general the editing process should be fully defined (in terms of what is permissible), and this is not currently done in the paper.
* The experimental results in Section 5 and as discussed in the paper overstate the superiority of the proposed method, especially with respect to Melodia. In reality as can be seen from Table 1, Melodia is ranked 1st across several metrics and datasets. The discussion in section 5 would need to be radically revised to reflect this.
* As can be seen from Table 2, even a simple change of the off-the-shelf source separation method used in the proposed system can lead to a drastic reduction in system performance. This poses the question whether the underlying Polyphonia model is indeed competitive, or whether most gains are to be made by simply using a reliable source separation method with a simpler editing component. Discussion related to these results also needs to be revised accordingly.

---

> ### Author Rebuttal · Authors · 2026-03-31
>
> Thank you for your time and effort in reviewing our research. We sincerely appreciate  your recognition of the good motivation and technical soundness of our method. Below, we provide responses to all the points raised.
>
> > Weakness1: Standardization of Terminology
>
> **Answer:** We agree that aligning with established MIR literature is essential for clarity. We will thoroughly standardize our terminology in the revision:
>
> - As suggested, we will explicitly rename our core task from "context-aware intra-stem editing" to "zero-shot timbre transfer in polyphonic music."
> - We will systematically replace non-standard terms to strictly match existing literature (e.g., updating "target ineffectiveness" to "target misalignment").
>
> > Weakness2: Early Signposting of Source Separation
>
> **Answer**: Thank you for pointing out this. In the revised manuscript, we will explicitly signpost the integration and critical role of BSS models in the Abstract and Introduction to ensure the system's operational dependency is transparent from the very beginning.
>
> > Weakneas3: Clarification on Coherence and Editing Permissibility
>
> **Answer**: We sincerely thank the reviewer for the insightful question regarding the editing process and coherence. We respectfully clarify that while pitch preservation prevents basic melodic clashes, it is insufficient for achieving overall coherence.
>
> - The **Overall Coherence (SongEval [1])** evaluates the complete perceptual and dynamic unity of the full track, which extends significantly beyond mere pitch accuracy.
> - In a context-blind "separation-editing-remixing" baseline, standard generative models arbitrarily hallucinate micro-timing, dynamics, and spatial acoustics for the isolated stem. Because these editing models **lack structural and contextual conditioning**, forcibly remixing the stem results in severe aesthetic disconnects.
> - In contrast, Polyphonia employs Attention Calibration via IRM. This ensures the newly transferred timbre dynamically interlocks with the background's acoustic groove, maintaining contextual harmony.
>
> To eliminate ambiguity, we will revise Section 4.1.2 to explicitly define our constraints: frameworks must perform zero-shot timbre transfer that strictly preserves the original pitch and rhythmic structure, while adaptively modifying timbre, dynamics, and articulation to acoustically blend with the polyphonic mixture.
>
> > Weakness4: Quantitative Discussion
>
> **Answer:** Thank you for the meticulous observation. We respectfully clarify that Melodia's top rankings on specific structural metrics stem from a methodological artifact rather than superior editing capability. Section 5 will be radically revised to include the following analysis:
>
> - Melodia aggressively preserving source features and frequently failing to execute actual timbre transfer (visually evident in Figure 1).
> - Consequently, its outputs retain extreme similarity to the source, artificially inflating structural metrics (LPAPS, KAD) without fulfilling the editing task.
> - Melodia struggles to lock fine-grained pitch envelopes, demonstrated by its lower CQT1-PCC score on the MUSDB18 dataset.
> - In contrast, Polyphonia successfully achieves zero-shot semantic editing while rigidly maintaining the structural envelope.
>
> > Weakness5: Dependency on BSS
>
> **Answer:** We respectfully clarify a misinterpretation of Table 2: the performance drop stems from removing our IRM module or reverting to a baseline, not from downgrading the BSS model. We will incorporate the following analysis into Section 5.4:
>
> - As demonstrated in Table 2, replacing the SOTA Demucs model with a simple, training-free "Naive DSP" results in only a marginal decrease in editing quality.
> - Crucially, Polyphonia with the lowest-tier Naive DSP still substantially outperforms the best BSS paired with a Sep-Remix baseline (e.g., CLAP: 0.432 vs. 0.334). This proves our IRM-based attention calibration—not the BSS backbone—drives the performance.
> - Because the Naive DSP relies exclusively on fundamental physical filtering rather than neural pre-training, it establishes a solid performance floor. This guarantees that Polyphonia remains highly robust and effective even in low-resource or novel-genre scenarios where modern neural BSS models might fail.
>
> ||Method|CLAP↑|CQT1-PCC↑|LPAPS↓|FAD↓|KAD↓|
> |-|-|-|-|-|-|-|
> |w/ IRM|Polyphonia+Demucs+$G_{X_0}$|0.437|0.547|4.096|0.949|0.695|
> ||Polyphonia+Unmix+$G_{X_0}$|0.435|0.541|4.173|0.953|0.696|
> ||Polyphonia+Naive+$G_{X_0}$|0.432|0.529|4.177|0.959|0.703|
> |w/o IRM|Polyphonia+Demucs+$G_{norm}$|0.413|0.459|4.194|1.229|0.728|
> ||Melodia (Sep-Remix)|0.334|0.692|2.937|0.623|0.510|
> ||DDPM (Sep-Remix)|0.330|0.532|3.887|0.869|0.648|
>
> **Reference:** [1] Yao, Jixun, et al. "Songeval: A benchmark dataset for song aesthetics evaluation." arXiv:2505.10793 (2025).

---

> > ### Author Rebuttal · Reviewer_hMfC · 2026-03-31
> >
> > The authors have successfully addressed my comments. Noting that the first comment might require a complete re-read of the manuscript as to ensure that terms have been adequately defined.

---

> > > ### Author Response · Authors · 2026-04-01
> > >
> > > We sincerely thank you for your meticulous and insightful feedback. Following your recommendation, we have completely re-read the entire manuscript and revised the terminology throughout the paper to ensure accurate alignment with standard literature.
> > >
> > > **Main Revisions**
> > >
> > > 1. **Title Update**
> > >
> > > We updated the title of our paper from "Polyphonia: Training-Free Context-Aware Music Editing with Acoustic-Informed Attention Calibration" to **"Polyphonia: Zero-Shot Timbre Transfer in Polyphonic Music with Acoustic-Informed Attention Calibration"** to better reflect the core methodology.
> > >
> > > **Corresponding revisions in paper:**
> > >
> > > - l. 33-34 -- context-aware music editing tasks → timbre transfer tasks in polyphonic music
> > >
> > > 2. **Task Definition Refinement**
> > >
> > > We refined our task definition from "context-aware intra-stem editing" to **"stem-specific timbre transfer"**. The revised definition (l. 34-45 in the Introduction) now reads:
> > >
> > > > In professional workflows, text-driven music editing generally addresses two levels of manipulation: global track arrangement (e.g., adding or removing a specific instrument) and stem-specific modification within a dense mixture. Within the latter, *stem-specific timbre transfer* represents a critical yet challenging task: surgically altering the timbre of a target stem (e.g., transforming a vocals track into a violin) while strictly preserving the acoustic integrity of its original pitch contours, rhythmic structure and all non-target stems (e.g., the co-occurring melody and accompaniment). This task poses a dual challenge: the newly transferred timbre must accurately reflect the semantic prompt while seamlessly blending with the rigidly preserved non-target stems.
> > >
> > > **Corresponding revisions in paper:**
> > >
> > > - l. 15, 39, 67, 84-85, 108, 148, 286, 821, 1063, 1108, 1171 -- context-aware editing / context-aware intra-stem editing  → stem-specific timbre transfer
> > > - l. 30-31, 103, 215-216, 252, 705 -- background context → non-target stems
> > > - l. 65 -- “to-be-preserved” context  → “to-be-preserved” non-target stems
> > > - Fig.1 & l. 27 -- Context Locking → Non-Target Preservation
> > > - l. 141, 156, 248, 705 -- background context / context  / context stem $S_\text{con}$  →  non-target mixture $S_\text{con}$
> > > - l. 220 -- the estimated target stem $ \tilde{S} _ \text{tgt} $ and context stem $ \tilde{S} _ \text{con} $ → ... non-target mixture $\tilde{S}_\text{con}$
> > > - l.256 -- context estimate $ \tilde{S} _ \text{con} $ → non-target estimate $ \tilde{S} _ \text{con} $
> > > - eq. 7 & l. 218 Context Preservation → Non-Target Preservation
> > > - l. 226 -- the acoustic environment of the context  → the acoustic environment of the non-target stems
> > > - l. 708 -- the context waveform → the non-target waveform
> > > - l. 191 -- with both target and context keys → with both target and non-target keys
> > > - l. 37, 205, 830, 1065 -- background integrity / background preservation → non-target integrity
> > > - l. 45, 1083 -- preserved background → preserved non-target stems
> > > - l. 416, 430 -- the background → the non-target stems
> > >
> > > 3. **Section Heading Update**
> > >
> > > We renamed the title of Section 4.1.2 from "External Acoustic Anchoring" to **"External Acoustic Prior"** to more accurately align with the definition of $G$ established in the paper.
> > >
> > > **Corresponding revisions in paper:**
> > >
> > > - l. 68 -- acoustic anchoring → acoustic prior
> > > - l. 187, 204 -- anchor the target alignment → enforce the target alignment
> > > - l. 193 -- unanchored guidance → unconstrained guidance
> > > - l. 209 -- External Acoustic Anchoring → External Acoustic Prior
> > > - l. 416 -- anchoring down layers → conditioning the down layers
> > > - I. 426 -- anchors the diffusion process → guides the diffusion process
> > > - l. 853 -- optimal anchor → optimal configuration
> > > - l. 1193 -- reference anchor → reference standard
> > >
> > > **Other Minor Revisions**
> > > - Fig.1 & l. 30, 196 -- Target Ineffectiveness → Target Misalignment
> > > - l. 22, 53, 198 -- semantic intent → semantic features of target stems
> > > - Fig.1 & l.26 -- Editing Leakage → Non-target Distortion
> > > - l. 60 -- internal feature locking → internal feature preservation
> > > - l. 61 -- lock entangled features → preserve entangled features
> > > - l. 103 -- strictly locking the background → strictly preserving the background
> > > - I. 131 -- vocal profile → vocal spectral envelope
> > > - I. 193, 202, 204 -- target spectral profile → target spectral envelope
> > > - I. 294 -- source’s sparsity profile → source’s sparsity pattern
> > > - l. 817-818 -- resolving the target leakage issues →  resolving the non-target distortion issues
> > > - l. 191-192 -- leading to the semantic leakage visualization → leading to the non-target distortion visualization

---

### Official Review · Reviewer_8mTn · 2026-03-11

**Soundness:** 3
**Presentation:** 3
**Significance:** 2
**Originality:** 2
**Overall Recommendation:** 3
**Confidence:** 3

**Summary:**

This paper proposes Polyphonia, a training-free music editing framework with context awareness. The authors leverage a probabilistic acoustic prior to calibrate attention and identify the boundaries, ensuring the preservation of background context and semantics. The authors also propose a standard prompt set -- PolyEvalPrompts for this task. Both objective and subjective experimental results demonstrate the effectiveness of Polyphonia.

**Compliance With Llm Reviewing Policy:**

Affirmed.

**Final Justification:**

I commend the authors for correcting their claims to 'Zero-Shot Timbre Transfer.' However, this clarification confirms that the task itself is overly constrained. A framework that fundamentally cannot handle structural or cross-articulation edits offers limited methodological significance to the broader machine learning community, leading me to maintain my Weak Reject.

**Key Questions For Authors:**

1. The authors use Qwen-Audio to analyze the source stems and build the PolyEvalPrompts benchmark. But in the real world, it is hard to request users to provide such precise prompts. Most users without musical expertise tend to give prompts that are sparse, ambiguous, or musically inaccurate. So it's worth discussing how robust the Polyphonia is when the user's prompt misaligns with the actual acoustic content of the source?
2. The title and abstract both claim that the proposed method is targeted for music editing. However, the Ideal Ratio Mask locks the spatial-spectral envelope, which makes it impossible to change the main melodies, chords, and rhythmic structures. This sounds like not align with the general understanding of music editing. Can the authors justify this terminology, or clarify if the framework should be strictly defined as Zero-Shot Timbre Transfer?
3. When the scope of a task is relatively limited, exploring and discussing its potential practical significance and applicability can help substantiate the task's meaning.

**Limitations:**

yes

**Strengths And Weaknesses:**

Strengths:

1. The paper is well-written and the figures/tables are visually appealing and clear.

2. The authors have successfully identified a real pain point and effectively solved the issue of editing specific tracks without ruining the background accompaniment.

3. The authors conducted a comprehensive and multi-faceted evaluation. The proposed method objectively and subjectively (MOS) outperforms strong baselines, such as Melodia and SteerMusic in balancing target alignment and background preservation.

4. The authors' effort in building the prompt set may be beneficial for the community.

5. The audio samples on the demo page sound impressive.

Weaknesses:

1. Both the abstract and title claim that the proposed method is addressing music editing, yet the method itself is strictly confined to zero-shot timbre transfer. The core IRM mechanism locks the physical envelope and spectral boundaries, making it impossible to rewrite melodies, chords, or rhythmic grooves. Thus, the description of the task is somewhat overclaimed and the applicability of the proposed method is limited.

2. Forcing a sustained instrument into the transient acoustic envelope of a percussive instrument will inevitably cause artifacts. The authors avoided these failure cases by restricting the PolyEvalPrompts dataset to only cover the change of functionally similar instruments. However, if the task is narrowed to the timbre swap of functionally similar instruments, the scope of application will be further compressed.

3. Initially, the paper points out that vanilla cross-attention tends to fail at disentangling dense music mixtures. In contrast, when discussing the "Other" stem in the appendix, the authors suddenly claim that the attention mechanism's semantic selectivity is sufficient on its own. This somewhat contradicts the core motivation.

4. According to the method details, the framework's performance is fundamentally limited by the capabilities of the off-the-shelf source separation model.

---

> ### Author Rebuttal · Authors · 2026-03-31
>
> Thanks very much for your careful feedbacks and recognition of good motivation, grounded method and impressive results of our work.
> > weakness1&question2: definition of task scope
>
> **Answer:** Thank you for their precise insights regarding the scope of our work. We agree that "music editing" is a broad umbrella term that inherently includes rewriting melodies, chords, and rhythmic structures, and we will refine our claims accordingly to prevent overclaiming:
>
> - **Terminology Correction:** Our framework is indeed specifically scoped to "Zero-Shot Timbre Transfer" rather than general music editing. We will revise the title, abstract, and introduction to explicitly reflect this precise terminology.
> - **Intentional Constraint via IRM:** Locking the spatial-spectral envelope via the IRM is a deliberate architectural design, not an unintended limitation. It explicitly acts as a coarse acoustic boundary to guarantee background preservation
> - **Optimal Trade-off:** While this mechanism inherently restricts the rewriting of rhythmic or melodic structures, this trade-off is necessary to resolve severe boundary leakage in dense mixtures. It provides a highly reliable solution specifically for stem-level timbre transfer.
>
> > weakness2&question3: application scope discussion
>
> **Answer:** Thank you for the insightful comments on the practical scope of our work and the dataset design. We clarify that our task definition and dataset construction are strictly driven by real-world music production logic:
>
> - The core value of *Zero-Shot Timbre Transfer in Polyphonic Music* lies in high-frequency professional workflows, such as rapid track prototyping and seamless arrangement refinement. In these scenarios, producers require precise timbre modification while strictly preserving the rhythmic groove of the background accompaniment.
> - As you correctly noted, *forcing a sustained envelope onto a transient percussive instrument inherently causes acoustic artifacts*. In real-world production, if a user wishes to preserve the original timing and groove, they naturally swap stems within functionally compatible families. Forcing a **cross-functional swap** (e.g., strings to a kick drum) while locking the original envelope is **musically illogical**.
> - Therefore, restricting `PolyEvalPrompts`  to functionally compatible swaps does not "artificially compress" the model's scope to avoid failure cases. Rather, it deliberately **aligns the benchmark with realistic, high-value industry use cases** to evaluate true practical utility.
>
> > weakness3: Clarification on CA for the "Other" Stem
>
> **Answer:** We acknowledge that "sufficient on its own" in the appendix is misleading. However, we would like to clarify that our underlying mechanism remains entirely consistent with our core motivation.
>
> - As stated in our core motivation, vanilla cross-attention fails to isolate targets in the global, dense mixture due to severe spectral interference.
> - **Reduced Search Space:** When handling stem within the "Other" stem, we absolutely do not apply vanilla CA directly to the global mixture, but the reduced search space which the background energy is filter out by IRM.
> - *Table: Performance Breakdown by Stem Type (MScK)* shows that CA can function effectively to resolve remaining local ambiguities.
>
> To eliminate any ambiguity, we will revise the phrasing to state that "cross-attention functions effectively within the reduced search space provided by the IRM", removing the misleading phrase "sufficient on its own."
> > weakness4: Dependency on BSS
>
> **Answer:** Please refer to **Reviewer hMfC weakness 5**.
>
> > question1: Robustness to user prompts
>
> **Answer:** To evaluate robustness against sparse descriptions, we constructed a "Sparse Prompt" set by stripping all detailed adjectives from the original prompts (e.g., simplifying "A fast expressive violin..." to merely "A song with violin, bass, melody, and drums").
> While the target alignment exhibits an expected drop due to the lack of fine-grained textual details, Polyphonia (CLAP: 0.322) still significantly outperforms all baselines equipped with full prompts (e.g., Melodia's CLAP: 0.296) (see table). This confirms that Polyphonia remains highly robust and effective for non-expert users.
>
> Table: Ablation study of prompt quality on MUSDB18-HQ
> |Type|CLAP↑|LPAPS↓|CQT1_PCC↑|KAD↓|FAD↓|
> |-|-|-|-|-|-|
> |Melodia|0.296|3.893|0.363|0.495|0.655|
> |SteerMusic|0.255|4.105|0.383|0.497|0.747|
> |MusicGen|0.295|6.600|0.003|0.840|1.374|
> |Polyphonia|0.342|4.426|0.371|0.645|0.868|
> |Polyphonia+Sparse Prompt|0.322|4.372|0.363|0.629|0.863|

---

> > ### Author Rebuttal · Reviewer_8mTn · 2026-04-02
> >
> > I appreciate the authors' honest, prompt rebuttal. The decision to explicitly re-scope the task to "Zero-Shot Timbre Transfer" and the additional ablation on "Sparse Prompts" are well-received and address my initial concerns regarding overclaiming and prompt robustness.
> > However, I am selecting option (c) and maintaining my original score of Weak Reject. Addressing the framework's strict reliance on the Ideal Ratio Mask (IRM) to lock the spatial-spectral envelope—which inherently prevents cross-articulation edits and structural generation—would require a significant architectural update to decouple the generation from the source's physical constraints. This fundamental update goes beyond what can be resolved in a short rebuttal. I thank the authors for their hard work and encourage them to explore this direction in future iterations.

---

> > > ### Author Response · Authors · 2026-04-03
> > >
> > > **Dear Reviewer 8mTn,**
> > >
> > > Thanks for your feedback. We acknowledge your assessment and appreciate your recognition of our task re-scoping and prompt robustness ablations.
> > >
> > > However, we respectfully maintain a fundamental difference in perspective regarding the framework's reliance on the IRM and your assertion that an architectural update is required to "decouple the generation from the source's physical constraints".
> > >
> > > As explicitly clarified in our re-scoping to **Zero-Shot Timbre Transfer**, locking the spatial-spectral envelope is not an architectural limitation awaiting a future update; it is the deliberate core mechanism designed to solve severe boundary leakage in dense polyphonic mixtures. In professional timbre transfer workflows, preserving the exact rhythmic and spectral envelope of the source tracks is the definitive requirement.
> > >
> > > Decoupling the generation to allow for cross-articulation or structural edits might destroy the strict acoustic boundaries, thereby reintroducing the very spectral interference and non-target stems' degradation that our method successfully eliminates. Evaluating a specialized, high-fidelity timbre transfer framework through the lens of general structural generation diverges from the established scope and practical utility of this work.
> > >
> > > We firmly stand by **our architectural design as a complete and robust solution for stem-specific timbre transfer**. We thank you again for your time and the rigorous discussion.

---

### Official Review · Reviewer_4LV3 · 2026-03-12

**Soundness:** 3
**Presentation:** 3
**Significance:** 3
**Originality:** 2
**Overall Recommendation:** 4
**Confidence:** 2

**Summary:**

This paper targets precise context aware intra stem editing of music signals. Breaking away from previous approaches, towards improved semantic-acoustic alignment, this paper uses acoustic anchoring for better stem localization and calibration of the attention masks. A prior mask (IRM) for the target stem is first obtained through an additional blind source separation (BSS) stage and this prior mask is then used later for attention calibration. The paper claims that this acoustic anchoring, at the cost of an additional BSS stage, achieves precise attention calibration needed for stem editing compared to baseline approaches.

**Compliance With Llm Reviewing Policy:**

Affirmed.

**Final Justification:**

While authors' response clarified some of my concerns, I still feel that the paper's contributions are limited. So I maintain my score as weak accept.

**Key Questions For Authors:**

1. The testing datasets contain limited number of stems. When the number of stems increase, does the BSS stage becomes a bottleneck? It will be good to see an ablation on the performance of the approach over varying number of stem tracks.

**Limitations:**

yes

**Strengths And Weaknesses:**

### Strengths
* The paper is well motivated and easy to follow.
* The method section does a good job of motivating the design choices.
* Overall, this paper is interesting, technically sound and tackles a very relevant application.
* Extensive ablation highlights importance of different modules of the proposed approach.

### Weaknesses

* Contribution is limited. The only detour from the previous approaches seems to be the use of a prior mask from the separated stem obtained through a preprocessing step. This introduces an extra preprocessing stage and the gain it brings is marginal (in Table 1 ) from melodia and steerMusic while increasing more vRAM and computation time (Table 3).
* The paper introduces a bunch of engineering in the method section to better calibrate the attention editing. But experiments are limited to only AudioLDM2 model. Results on more baselines are needed to effectively validate the efficacy of the proposed approach.
* Explanation/discussion on the test sets are limited. More details on types of stems, number of stems per audio etc would be helpful judge the merit of the proposed method.

---

> ### Author Rebuttal · Authors · 2026-03-31
>
> Thanks for your positive feedback and valuable questions. We greatly appreciate your recognition of the great motivation and technical soundness of our method.
> > Weakness: Limited Contribution
>
> **Answer:** We respectfully clarify that our contribution extends significantly beyond a preprocessing step, and the performance gains are substantial rather than marginal:
>
> - **Architectural Modification:** We do not simply apply a mask to audio. We first formulate BSS output into IRM, which is a carefully selected prior suits for attention map, and then use it to anchor attention via novel Attention Calibration mechanism (Reviewer MScK).
> - **Resolving Spectral Interference:** Existing methods rely on internal attention maps, which fail to isolate targets in global dense mixtures due to severe spectral interference. Our approach directly resolves this structural flaw by mathematically coupling objective acoustic boundaries with text-driven semantic attention.
> - **Substantial Performance Gains:** Achieving a 15.5% relative improvement in target alignment (CLAP) while simultaneously securing state-of-the-art results on composite balance metrics (AMB/ASB)  represents a significant technical advancement in context-aware music editing, not a marginal gain.
>
> > Weakness: vRAM and computation time overhead
>
> **Answer:** Thanks for raising the crucial discussion on computational efficiency.
>
> First, we would like to highlight a critical correction. The peak VRAM for Polyphonia reported in the original Appendix (Table 3) was artificially inflated due to a bug that redundantly stored unused tensors during calibration. We have fixed this memory leak prior to the rebuttal **without affecting any generation results or inference time**. Results demonstrate that the BSS prior achieves superior editing performance **without drastically increasing computation time or memory requirements across audio length**.
>
> Table: Efficiency Comparison across audio length
> |Method|Audio Length|Pre-process(s)|Inversion(s)|Generation(s)|Total Time(s)|Peak VRAM(GB)|
> |-|-|-|-|-|-|-|
> |SDEdit|5s|-|0.48|7.86|8.34|4.39|
> ||10s|-|0.50|9.50|10.00|4.49|
> ||15s|-|0.56|12.57|13.13|4.60|
> |MusicGen|5s|-|-|8.89|8.89|4.19|
> ||10s|-|-|17.93|17.93|4.21|
> ||15s|-|-|23.67|23.67|4.28|
> |DDIM|5s|-|9.32|9.43|18.75|4.41|
> ||10s|-|9.81|9.35|19.16|4.50|
> ||15s|-|10.87|10.74|21.61|4.57|
> |DDPM|5s|-|9.79|9.61|19.40|4.42|
> ||10s|-|10.01|9.59|19.60|4.52|
> ||15s|-|10.38|10.18|20.56|4.62|
> |Melodia|5s|-|9.89|9.62|19.51|6.92|
> ||10s|-|10.13|9.88|20.01|9.62|
> ||15s|-|11.35|10.82|22.17|12.93|
> |MusicMagus|5s|-|43.56|34.88|78.44|6.95|
> ||10s|-|44.33|36.09|80.42|8.69|
> ||15s|-|49.75|46.16|95.91|11.47|
> |SteerMusic|5s|-|4.26|48.48|52.74|4.51|
> ||10s|-|4.26|51.97|56.23|4.64|
> ||15s|-|4.26|73.12|78.38|4.77|
> |Polyphonia (Ours)|5s|1.27|10.79|10.54|22.60|5.30|
> ||10s|1.52|11.53|11.33|24.38|8.05|
> ||15s|1.61|10.88|12.02|23.90|13.68|
>
> > Weakness: Generalizability
>
> **Answer:** To demonstrate that Polyphonia is model-agnostic, we integrated it into **Tango[1]**, another prominent text-to-audio diffusion model. As shown below, applying Polyphonia improves all metrics compared to the vanilla inversion-denoising baseline.
>
> Table: Quantitative results on MUSDB18-HQ.
> |Method|CLAP↑|CQT1-PCC↑|LPAPS↓|FAD↓|KAD↓|
> |-|-|-|-|-|-|
> |Tango|0.302|0.365|5.059|0.964|0.706|
> |Tango+Polyphonia|0.306|0.491|4.530|0.704|0.577|
> > Weakness&Question 1: test set details and ablation study with number of stems
>
> **Answer:** We fully agree that providing more granular details about the test sets helps contextualize the robustness of our method:
>
> *MUSDB18:* 99 unique stem types, song type distribution as follow:
>
> |Song type|Count|
> |-|-|
> |1-stem|1|
> |2-stem|1|
> |3-stem|13|
> |4-stem|23|
> |5-stem|10|
> |6-stem|2|
> ||50|
>
> *MusicDelta:* 46 unique stem types, song type distribution as follow:
>
> |Song-type|Count|
> |-|-|
> |3-stem|12|
> |4-stem|14|
> |5-stem|1|
> |9-stem|1|
> ||28|
>
> We evaluated Polyphonia across varying stem counts to analyze potential BSS bottlenecks (see table).
>
> Table: Ablation over varying number of stem tracks
>
> |Dataset|Stem-number|CLAP↑|CQT1-PCC↑|LPAPS↓|
> |-|-|-|-|-|
> |MUSDB18|≤ 3|0.343|0.369|4.457|
> ||4|0.343|0.374|4.439|
> ||≥ 5|0.341|0.367|4.367|
> |MusicDelta|≤ 3|0.433|0.507|4.185|
> ||4|0.444|0.587|4.006|
> ||≥ 5|0.412|0.507|4.186|
>
> - **Consistent in Typical Scenarios:** Performance remains stable in the MUSDB18 dataset, demonstrating robust generalization.
> - **Graceful Degradation at Limits:** In MusicDelta, a slight performance drop occurs only in extremely dense mixtures ($\ge 5$ tracks, up to 9). This confirms that while overwhelming track density forms a natural upper bound for BSS-guided methods, Polyphonia degrades gracefully without catastrophic failure.
>
> **References:** [1]Majumder, Navonil, et al. "Tango 2: Aligning diffusion-based text-to-audio generations through direct preference optimization." *Proceedings of the 32nd ACM International Conference on Multimedia*. 2024.

---

> > ### Author Rebuttal · Reviewer_4LV3 · 2026-04-03
> >
> > I thank the authors for their detailed rebuttal and the effort invested in addressing my concerns regarding memory/computation overhead, generalizability, and test set details. The new results effectively resolve these points.
> >
> > However, my primary concern regarding the paper's contribution remains. Utilizing BSS for IRM extraction is a straightforward application and is not novel. While it is interesting to observe the method's robustness to the BSS stage, the overall contribution still feels limited. And since improving contributions is beyond the score of this rebuttal, I will maintain my current score.

---

> > > ### Author Response · Authors · 2026-04-03
> > >
> > > **Dear Reviewer 4LV3,**
> > >
> > > Thanks for your feedback. We appreciate your confirmation that the additional results provided in our rebuttal have effectively resolved your initial concerns regarding memory/computation overhead, generalizability, and test set details.
> > >
> > > However, we respectfully maintain strong reservations regarding your assessment of our technical contribution and must reaffirm our stance on the novelty of this work.
> > >
> > > Characterizing our framework as a "straightforward application" of BSS is reductive and fundamentally overlooks the architectural complexity of our proposed method. As explicitly detailed in our rebuttal, the primary innovation is **not** the isolated extraction of the IRM. Rather, the core contribution is the **Attention Calibration** mechanism. This novel mechanism mathematically couples the objective acoustic prior with the text-driven semantic attention of diffusion models to directly resolve **Spectral Interference** in global dense mixtures—a persistent structural flaw that existing methods relying on internal attention maps completely fail to address.
> > >
> > > Evaluating Polyphonia solely by dismissing its preprocessing component fails to acknowledge the synergistic innovation of the entire pipeline. Securing a 15.5% relative improvement in target alignment (CLAP) while simultaneously achieving SOTA performance on composite balance metrics (AMB/ASB) provides empirical proof that our method is a substantial technical advancement, rather than a marginal or straightforward application.
> > >
> > > We stand firmly by the technical soundness, significant novelty, and practical impact of our proposed architecture. We thank you for your time during the review process.

---

### Official Review · Reviewer_MScK · 2026-03-13

**Soundness:** 2
**Presentation:** 2
**Significance:** 3
**Originality:** 3
**Overall Recommendation:** 4
**Confidence:** 4

**Summary:**

Polyphonia is a training-free framework for context-aware intra-stem music editing. It addresses semantic-acoustic misalignment in diffusion-based editing by injecting an Ideal Ratio Mask (IRM), derived from blind source separation, into attention layers via two mechanisms: Source Interpolation (pre-softmax blending) and Acoustic Modulation (spatio-textual cross-attention bias). The authors also contribute PolyEvalPrompts, a benchmark of 1,170 editing tasks across MUSDB18-HQ and MusicDelta, and report consistent improvements in target alignment and background preservation over strong training-free baselines.

**Compliance With Llm Reviewing Policy:**

Affirmed.

**Final Justification:**

I really appreciate the serious work conducted by the authors in addressing my concerns successfully.

**Key Questions For Authors:**

1. When the target instrument falls within the "other" stem (e.g., electric guitar, violin), how exactly is the GIRM computed — do you use the full "other" mask, and if so, how do you quantify the precision loss relative to a dedicated stem? Have you benchmarked edits within "other" separately from vocals-class edits?
2. In the Source Interpolation mechanism, how is the gate G broadcast over the L_z × L_z self-attention energy matrix — row-wise per query position, column-wise per key position, or as an outer product? How sensitive are your metrics to this choice?
3. Have you explored per-layer or time-step schedules for the modulation strength λ, or negative bias applied to non-target tokens to actively suppress leakage, rather than only boosting target token attention?
4. The paper frames Polyphonia as "training-free," yet it depends on a pretrained BSS model. How much does editing quality degrade as BSS quality decreases, and at what BSS performance floor does the method become inferior to the non-BSS baselines? Could this dependency be a practical barrier in low-resource or novel-genre settings?
5. What are the end-to-end wall-clock times and peak GPU memory requirements compared with the reported baselines, and how does this scale with audio duration? Without these numbers, it is difficult to assess whether the gains justify the added BSS overhead in production settings.

**Limitations:**

Yes.

**Strengths And Weaknesses:**

**Soundness**

Pro: IRM as a soft acoustic prior is well-grounded; ablations on prior type and separation model quality are thorough.
Cons: Broadcast semantics of the gate G in self-attention are underspecified; the PPAE baseline is excluded without a re-implementation or rigorous justification.

**Originality**

Injecting a separation-derived acoustic prior into pre-softmax attention is novel relative to existing gradient-steering and structural-reweighting baselines. However, IRM-guided attention shows conceptual overlap with audio-visual segmentation work that is not adequately cited; the novelty claim would benefit from sharper differentiation.

**Presentation**
Construction of the token mask m_text for multi-word and ambiguous targets is underspecified; the mapping from edit prompt to BSS stem is not fully detailed, especially within the broad "other" category.

---

> ### Author Rebuttal · Authors · 2026-03-31
>
> Thanks very much for your comments. We sincerely appreciate your recognition of the good soundness and novelty of our method.
> > Presentation2&Question1: issues of the "Other" Category
>
> **Answer:** Yes, when the target stem falls into the "Other" stem, we use the full "Other" mask as a coarse acoustic prior.
>
> Regarding precision loss, our system remains highly robust by:
>
> - **Ablation Evidence:** As shown in Table 2, Naive DSP IRM—which intentionally simulates severe precision loss—outperforms the baseline Demucs G_norm. This proves that as long as a safe, coarse search space is constructed, the cross-attention mechanism can successfully resolve the target.
> - **Quantitative Benchmark:** We benchmarked performance across different stem types. As shown below, performance within the "Other" category is highly competitive with "Vocals," confirming no severe precision degradation.
>
> Table: Performance Breakdown by Stem Type
> |Dataset|Stem Type|CLAP↑|LPAPS↓|CQT1_PCC↑|
> |-|-|-|-|-|
> |MUSDB18|Vocals|0.337|4.630|0.369|
> ||Others|0.339|4.355|0.363|
> |MusicDelta|Vocals|0.406|3.964|0.602|
> ||Others|0.433|4.129|0.526|
> > Soundness&Question2: G Broadcast in Self-Attention
>
> **Answer:** We apologize for absence of clarification of $G$ broadcasting mechanism  over the $L_z \times L_z$ self-attention. We use **row-wise per query position** to apply $G$.
>
> Regarding other approaches, we clarify that the broadcast dimension $Q$ is a strict mathematical requirement of Polyphonia, not a tunable hyperparameter.
>
> - **Row-wise per Query (Adopted):** Broadcasting row-wise applies the scalar $G_i$ uniformly across all Keys for the $i$-th Query. This ensures each token either adopts the current editing state ($E_{cur}$) or the cached background state ($E_{src}$), aligning with our editing goal.
> - **Column-wise per Key:** Forces a single Query to mix uncalibrated logits from different generation stages (inversion vs. editing). This internal scale mismatch mathematically shatters the Softmax normalization.
> - **Outer Product:** A 2D outer product mask only uses new features at the intersection of "target queries" and "target keys". When a target query needs to attend to the background environment, it is forced to regress to cached features.
>
> > Question3: λ schedule
>
> **Answer:**
>
> - **Time-Step Scheduling:** We explored time-step decay strategies for $\lambda$ (see table). While decay marginally improves structural preservation, constant schedule maximizes target alignment.
>
> Table: Ablation on $\lambda$ Time-Step Scheduling (max $\lambda=2.5$)
> |Schedule Type|CLAP↑|LPAPS↓|CQT1_PCC↑|KAD↓|FAD↓|
> |-|-|-|-|-|-|
> |Constant|0.437|4.096|0.547|0.695|0.949|
> |Linear Decay|0.435|4.083|0.549|0.686|0.942|
> |Cosine Decay|0.435|4.081|0.549|0.686|0.941|
>
> - **Per-Layer Scheduling:** Our ablation(Fig. 6) isolates modulation to the down-sampling layers. We maintain a uniform $\lambda$ across layers because the acoustic prior $G$ represents an objective physical energy boundary. Varying $\lambda$ per layer would mathematically imply that the location of stem shifts at different abstraction levels
> - **Negative Bias:** Applying a positive bias to the target tokens pre-softmax inherently achieves this exact negative suppression effect. We chose to employ only the positive bias to reduce the hyperparameter search space.
>
> > Soundness: PPAE
>
> **Answer:** Exclusion of PPAE was a deliberate choice for follow reasons:
>
> - **Theoretical & Empirical Limits (Sec 4.1.1 & Fig. 2):** PPAE relies on internal attention maps for masking, making PPAE structurally unsuited for this task.
> - **Domain Mismatch:** PPAE is optimized for general Audio Events, whereas our benchmark targets highly entangled, multi-track music.
> - **Fairness in Reproduction:** We reproduced PPAE and observed the exact target leakage predicted by our analysis. However, without official code or hyperparameters, we excluded these degraded results to avoid an unoptimized and potentially unfair comparison.
>
> > Presentation: m_text
>
> **Answer:** $m_{text}$ is formally defined as a binary indicator in Equation 12. To explicitly address multi-word targets, $m_{text}$ applies this by assigning $1$ to all constituent target token indices and $0$ otherwise, thereby isolating all relevant attention columns along the Key dimension.
>
> > Originality: Audio-Visual Segmentation(AVS)
>
> **Answer:** While AVS uses audio signals to ground visual pixels, Polyphonia addresses a fundamentally different generative challenge within the music domain. Our IRM-guided attention does not merely align feature representations; instead, it leverages IRM as a mathematical constraint to modulate the attention dynamics of a diffusion model. Therefore, our novelty lies in repurposing acoustic priors as dynamic structural boundaries for music editing, rather than serving as cross-modal alignment targets.
>
> > Question4: Dependency on BSS
>
> Please refer to **Reviewer hMfC Weakness 5**.
>
> > Question5: Efficiency
>
> Please refer to **Reviewer 4LV3 Weakness VRAM**.

---

> > ### Author Rebuttal · Reviewer_MScK · 2026-04-03
> >
> > The authors resolved my main concerns. Ensuring these are appropriately included in the paper requires a reread.

---

> > > ### Author Response · Authors · 2026-04-04
> > >
> > > Thanks for your positive feedback. We sincerely appreciate your confirmation that the additional results have effectively resolved your initial concerns.
> > >
> > > Following your recommendation, to ensure that all the added results and analysis are appropriately included in the paper, we have completely re-read the entire manuscript and revised the correlated sections throughout the paper:
> > >
> > > **1. Performance and Robustness of the "Other" Stem**
> > >
> > > - In **Section 5.4**. We **included the new quantitative benchmark table "Performance Breakdown by Stem Type" across the MUSDB18 and MusicDelta datasets**. And we explicitly states the ablation evidence proving that the cross-attention mechanism successfully resolves the target without severe precision degradation, even when utilizing the coarse "Other" mask.
> > > - In **Appendix C.2**. We clarified the cross-attention (CA) mechanism for the "Other" stem. To eliminate any ambiguity, we explicitly stated that "cross-attention functions effectively within the reduced search space provided by the IRM." We emphasized that while vanilla CA fails to isolate targets in a global dense mixture due to spectral interference, it successfully resolves remaining local ambiguities once the non-target energy is filtered out.
> > >
> > > **2. Clarification of the $G$ Broadcast Mechanism in Self-Attention**
> > >
> > > * In **Section 4.2.1**, we explicitly states that we use the **row-wise per query** broadcasting mechanism to apply $G$ over self-attention after introducing eq.10, 11.
> > > * In **Appendix C.4**, we added the mathematical justification for why this specific broadcast dimension is a strict requirement for Polyphonia  and explicitly detailed why alternatives like column-wise or outer product masking mathematically fail to preserve the Softmax normalization and background features.
> > >
> > > **3. $\lambda$ Scheduling Ablation**
> > >
> > > * In **Appendix C.5**, we added statement: "Empirical observations in Appendix E suggested that time-dependent decay schedules for $\lambda$ yield marginal improvements in non-target integrity but fail to maximize target alignment compared to a fixed prior."
> > > * In **Appendix E**, we incorporated the new ablation table "Ablation on $\lambda$ Time-Step Scheduling" to empirically justify why a constant schedule maximizes target alignment. We also added the theoretical rationale explaining why varying the scalar per layer would mathematically imply shifting target locations at different abstraction levels, justifying our uniform application across down-sampling layers.
> > >
> > > **4. Discussion of PPAE**
> > >
> > > - In **Section 2.2**, we explicitly stated that "PPAE is optimized for general Audio Events, whereas timbre transfer in polyphonic music task targets highly entangled, multi-track music."
> > > - In **Appendix G**, we included the reason for the exclusion of PPAE: "We reproduced PPAE and observed the exact target leakage predicted by our analysis. However, without official code or hyperparameters, we excluded these degraded results to avoid an unoptimized and potentially unfair comparison."
> > >
> > > **5. Clarification of $m_\text{text}$**
> > >
> > > - In **Section 4.2.2**, immediately after defining the token mask $m_\text{text}$ (Equation 12), we explicitly added a pointer guiding readers to Appendix C.3 for comprehensive details on its tokenization logic and construction.
> > > - In **Appendix C.3**, we expanded the explanation of the binary token mask $m_\text{text}$. We explicitly clarified how it handles multi-word targets by functioning as a formal binary indicator (Equation 12).
> > >
> > > **6. Analysis of dependency on BSS**
> > >
> > > * In **Section 5.4**, we incorporated the comparative analysis demonstrating that Polyphonia with the lowest-tier Naive DSP model still substantially outperforms the best BSS paired with a Sep-Remix baseline. The text now clearly emphasizes that the performance drop shown in Table 2 is a result of removing the IRM module entirely, definitively proving that our acoustic prior—not the neural BSS backbone itself—is the primary driver of performance.
> > >
> > > **7. Corrected and Expanded Efficiency and VRAM Overhead Analysis**
> > >
> > > * In **Appendix D**, the old Table 3 has been replaced with the comprehensive new table demonstrating scaling across different audio lengths (5s, 10s, 15s). The text in Appendix D has been revised to note the correction of the VRAM caching bug, reflecting the accurate, significantly lower peak VRAM usage (e.g., ~8.05 GB for a 10s clip instead of the previously reported 13.22 GB), proving the method's computational viability.
> > >
> > > **8. Originality and Relationship with Audio-Visual Segmentation**
> > >
> > > In **Section 2**, we added a new subsection, **Section 2.3 (Audio-Guided Masking Paradigms)**, to explicitly discuss prior mask-based approaches and **clarify the fundamental distinctions between AVS and our framework**.
> > >
> > > Thank you again for helping us significantly strengthen the clarity and rigor of this paper.

---

### Decision · Program_Chairs · 2026-04-30

**Decision:**

Accept (regular)

**Comment:**

Polyphonia proposes a training-free, zero-shot music editing approach using a probabilistic acoustic prior (IRM) to guide semantic attention in a diffusion-based framework. Scores are 4, 4, 3, 4, reflecting a generally positive but not unambiguous reception.

Reviewers consistently praised the paper's clarity and ablation studies (MScK, 4LV3, hMfC), and the demo page samples were noted as compelling (8mTn). The IRM-as-soft-acoustic-prior framing was highlighted as the novel technical contribution (MScK).
The most substantive criticism, raised most forcefully by 8mTn and echoed by hMfC, concerns the scope of the claimed contribution. The paper frames itself as a general music editing system, but the IRM mechanism constrains it to timbre transfer: melody, chord, and rhythmic structure cannot be modified by design. 8mTn considers this overclaiming, and while the authors attempted a reframing toward "zero-shot timbre transfer" in the rebuttal, the reviewer maintains their score. hMfC similarly notes that the contribution appears overstated, particularly since Melodia outperforms the proposed method on objective metrics. 4LV3 independently reaches a similar conclusion — the core departure from prior work amounts to prepending a source-separation-derived prior mask, with marginal gains.

A related concern (8mTn, hMfC) is the hard dependency on off-the-shelf source separation: performance degrades substantially with weaker separators, which is a significant practical limitation that should be foregrounded. hMfC also flags non-standard terminology as an accessibility issue. 4LV3 notes that experiments are confined to AudioLDM2, leaving the generality of the approach unvalidated. MScK's original concerns about underspecified attention gating and unjustified baseline exclusion were addressed in the rebuttal to that reviewer's satisfaction, as were hMfC's concerns.

Recommendation: Weak accept. The paper is well-executed and the ablation work is thorough, but the contribution framing requires correction — the method should be clearly scoped as timbre transfer rather than general music editing. The source separation dependency should be prominently disclosed, and evaluation on additional base models would substantially strengthen the empirical case.